# Theory and Approximate Solvers for Branched Optimal Transport with Multiple Sources

**Peter Lippmann**    **Enrique Fita Sanmartín**    **Fred A. Hamprecht**
IWR at Heidelberg University, 69120 Heidelberg, Germany
{peter.lippmann, enrique.fita.sanmartin, fred.hamprecht}
@iwr.uni-heidelberg.de

## Abstract

Branched optimal transport (BOT) is a generalization of optimal transport in which transportation costs along an edge are subadditive. This subadditivity models an increase in transport efficiency when shipping mass along the same route, favoring branched transportation networks. We here study the NP-hard optimization of BOT networks connecting a finite number of sources and sinks in $\mathbb{R}^2$. First, we show how to efficiently find the best geometry of a BOT network for many sources and sinks, given a topology. Second, we argue that a topology with more than three edges meeting at a branching point is never optimal. Third, we show that the results obtained for the Euclidean plane generalize directly to optimal transportation networks on two-dimensional Riemannian manifolds. Finally, we present a simple but effective approximate BOT solver combining geometric optimization with a combinatorial optimization of the network topology.

## 1    Introduction

Optimal transport (OT) [27, 6, 23] stipulates transportation costs that increase linearly with the transported mass. However, in many systems of practical and theoretical interest, a *diminishing cost* property is more realistic: it is more economic to jointly transport two loads with nearby destinations along the same route. The optimal transportation networks under diminishing costs exhibit branching; and indeed, nature and societies are using branched networks, e.g. in blood circulation, gas supply or mail delivery. In this paper, we study the theory and practice of finding good or even optimal solutions in branched optimal transport (BOT).

More formally, we consider a finite set of *sources* $S$ with supplies $\mu_S > 0$ and *sinks* $T$ with demands $\mu_T > 0$, located at fixed positions $x_S$ and $x_T$ in $\mathbb{R}^2$. A possible transportation network is represented as a directed, edge-weighted graph $G(V, E)$ with nodes $V = S \cup T \cup B$. The edges $E \subset V \times V$ interconnect the terminals $S$ and $T$ with the help of a set of additional nodes $B$, so-called *branching points* (BPs), with coordinates $x_B$. The edge direction indicates the direction of mass flow. The edge weights, denoted by $m_e$, specify the absolute flows. Gilbert first proposed the BOT problem [9] in which the objective is to solve for

$$\underset{B,E,x_B,m_E}{\arg\min} \sum_{(i,j)\in E} m_{ij}^{\alpha} \|x_i - x_j\|_2 \text{, subject to} \tag{1}$$

$$\text{supply } \mu_s = \sum_k m_{sk} - \sum_k m_{ks} \text{ at each source } s,$$

$$\text{demand } \mu_t = \sum_k m_{kt} - \sum_k m_{tk} \text{ at each sink } t,$$

$$\text{conservation } \sum_k m_{kb} = \sum_k m_{bk} \text{ at each BP } b,$$

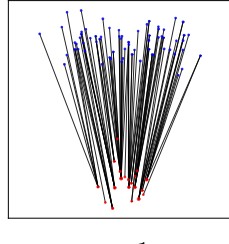
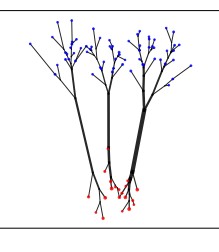
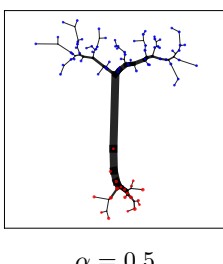
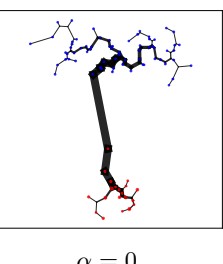

$\alpha = 1$          $\alpha = 0.95$          $\alpha = 0.5$          $\alpha = 0$

Optimal Transport          Euclidean Steiner Tree

Figure 1: Branched optimal transport (Eq. 1) interpolates between optimal transport and the Euclidean Steiner tree problem. On a toy example, shown are good BOT solutions (found by our approximate solver for $\alpha \neq 1$, see Sect. 6) for the same set of sources (red) and sinks (blue). The disk sizes indicate the demands and supplies, the edge widths the mass transported along each edge.

given a single parameter $\alpha \in [0, 1]$. The problem of BOT is interesting in that it combines combinatorial optimization (over $B$, $E$) with continuous optimization (over $x_B$, $m_E$).

For $\alpha = 1$, the BOT problem is the discrete version of the famous optimal transport problem for which optimal solutions can be found efficiently [6, 23]. However, due to the linearity of the cost function, OT solutions do not exhibit any branching but consist of straight lines between sources and sinks, see Fig. 1a. In contrast, for $\alpha \in [0, 1)$, the subadditivity of $m \mapsto m^\alpha$ reflects the increased efficiency of transporting loads together, i.e. $(m_1 + m_2)^\alpha < m_1^\alpha + m_2^\alpha$. Thus, for $\alpha \in [0, 1)$, BOT solutions show a branched structure, see Fig. 1b-d. Unlike OT, the optimization problem of BOT is NP-hard [11]. In the special case of $\alpha = 0$, BOT turns into the well-studied Euclidean Steiner tree problem (ESTP) [28, 13]. In the ESTP, the objective is to find the overall shortest network that interconnects all terminals (with the help of BPs), independently of the edge flows, since $m^0 = 1$. For different values of $\alpha$, BOT interpolates between these two optimization problems, see Fig. 1.

**Connection of BOT to machine learning.** Optimal transport has emerged as an important tool in machine learning [1, 5, 23]. BOT is a strict generalization, describing a more versatile concept and more challenging optimization problem.

BOT offers a mathematical formalism that is deceivingly simple (cf. Eq. (1)) and yet engenders non-trivial structure. Many machine learning problems such as tracking of divisible targets (computer vision), skeletonization (image analysis), trajectory inference (bioinformatics) come with input that is essentially continuous (images, distributions) and require structured output that is discrete, e.g. graphs. Arguably, this transition from continuous to discrete is one of the most interesting aspects (and an unsolved problem) in current machine learning research. It is also a problem that cannot be solved by a mere upscaling of standard deep learning architectures.

In addition, routing problems have become a popular problem to challenge machine learning and amortized optimization algorithms with difficult optimization problems [17, 3, 15]. Combining combinatorial and continuous optimization, BOT is a highly instructive target for new machine learning approaches. In Sect. 7 we address the generalization of BOT to higher-dimensional Euclidean space, particularly relevant for applications in data science.

In this paper, we make the following contributions: We generalize an existing method for constructing BOT solutions with optimal geometry to the case of multiple sources. Based on this generalization, we present an analytical and numerical scheme to rule out $n$-degree branchings with $n > 3$. Further, we demonstrate how to extend geometric and topological properties of optimal BOT solutions to two-dimensional Riemannian manifolds. Lastly, we propose a more practical numerical algorithm for the geometry optimization together with a simple but compelling heuristic, addressing the optimization of the BOT topology. To the best of our knowledge, no readily accessible code for finding BOT solutions is publicly available. By making our code available at `https://github.com/hci-unihd/BranchedOT` we hope to aid the evolution of the field.

## 2 Topology and geometry of BOT solutions

A BOT problem can be divided into the combinatorial optimization of the network topology, specified by the set of BPs $B$ and edges $E$ (see Sect. 1), and the geometric optimization of the BP positions $x_B$. Bernot et al. [2] showed that optimal BOT solutions can be assumed to be acyclic, which restricts the search for the optimal topology to trees. Given $n$ terminals, WLOG, the topology can be represented as a so-called *full tree topology*, which has $n-2$ BPs, each of degree three. Higher-degree branchings may effectively form during the geometry optimization if multiple BPs settle at the same position. A set of such BPs is referred to as *coupled* BP, cf. Fig. 2b. The union set of all neighbors of the individual BPs (not including the BPs themselves) is referred to as set of *effective neighbors*. Conversely, a BP configuration in which all BPs are uncoupled and located away from the terminals is called *non-degenerate*, see Fig. 2a.

The number of distinct full tree topologies interconnecting $n$ terminals is given by $(2n-5)!! = (2n-5) \cdot (2n-7) \cdot ... \cdot 3 \cdot 1$ and hence increases super-exponentially with the number of terminals [25]. Given 100 terminals, one would have to consider more than $10^{18}$ possible full tree topologies, making an exhaustive search computationally intractable already for problems of modest size. Fortunately, given a tree topology, the geometric optimization of the BP positions reduces to a convex optimization problem, as all edge flows $m_{ij}$ are already uniquely determined by the flow constraints in Eq. (1). The corresponding linear system can be solved in linear time by dynamic programming, called "elimination on leaves of a tree" in [26]. Since the Euclidean norm, like any norm, is convex, given a fixed tree topology, the cost function in Eq. (1) becomes a convex function of the BP positions. Together with the independence of the individual BPs, this implies the following lemma on the optimal substructure of BOT solutions (see App. C).

**Definition 2.1.** For a chosen topology $T$, a BOT solution is called a *relatively optimal solution* (ROS of $T$) if its BP configuration has minimal cost. The overall best BOT solution, given by the optimal topology together with its ROS, is called the *globally optimal solution* (GOS).

**Lemma 2.1.** *(a) For a given tree topology, a BOT solution is relatively optimal if and only if every (coupled) BP connects its (effective) neighbors at minimal cost. (b) In a globally optimal solution, every subsolution restricted to a connected subset of nodes solves its respective subproblem (cf. App. C) globally optimally.*

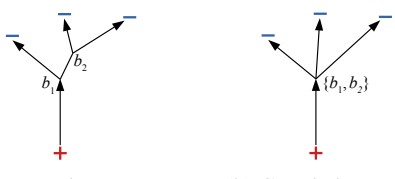

(a) Non-degenerate BP configuration

(b) Coupled BP with effective degree 4

Figure 2: Two BP configurations for the same full tree topology: A higher-degree branching may effectively form by coupling the two BPs at the same position.

## 3 Geometric optimization of BOT solutions

Although the BP optimization for a given tree topology is a convex problem, as argued above, it is non-trivial, since the objective function is not everywhere differentiable. Here, we present a principled geometric approach, which was first suggested by Gilbert in [9] and previously developed in the context of the ESTP [20]. More recently, this approach was discussed in the comprehensive work by Bernot et al. [2], where it was applied exclusively to BOT problems with a single source. A generalization to the case of multiple sources was posed as an open problem by the authors (see Problem 15.11), for which we give the solution in this section.

### 3.1 Geometric solution for one source and two sinks

Motivated by Lem. 2.1, we start by considering a single BP $b$ in isolation (cf. Fig. 3a), following [2]. Given a source at position[1] $a_0$ and two sinks at positions $a_1$ and $a_2$, we aim to find the optimal position $b^*$ for the BP connecting the three terminals, i.e., the minimizer of

$$\mathcal{C}(b) = m_1^\alpha |a_1 - b| + m_2^\alpha |a_2 - b| + (m_1 + m_2)^\alpha |a_0 - b|, \tag{2}$$

where $m_1$ and $m_2$ are the respective demands of the two sinks. Due to the convexity of $\mathcal{C}(b)$, the minimum must lie either at a stationary point at which $\nabla_b \mathcal{C} = 0$ or at a non-differentiable point,

---

[1] We will often use the node label, e.g., $a_0$, to denote also the position of the node, instead of writing $x_{a_0}$.

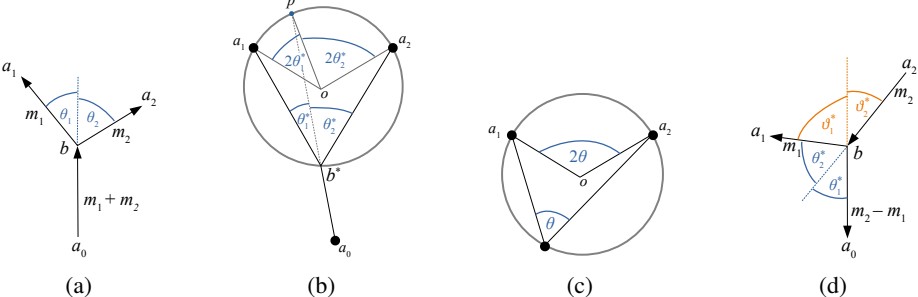

Figure 3: **(a)** Branching point $b$ connecting one source and two sinks with branching angles $\theta_1$ and $\theta_2$. **(b)** Construction of the optimal BP $b^*$ applying twice the central angle property illustrated in **(c)**. **(d)** shows the relation of the optimal branching angles $\theta_i^*$ and $\vartheta_i^*$, relevant for the case of asymmetric branchings (see Sect. 3.2).

where $b$ coincides with one of the $a_i$. Bernot et al. showed that the gradient is equal to zero if and only if the branching angles $\theta_i$, see Fig. 3a, are given by

$$\theta_1^* = \arccos\left(\frac{k^{2\alpha} + 1 - (1-k)^{2\alpha}}{2k^\alpha}\right) =: f(\alpha, k),$$

$$\theta_2^* = \arccos\left(\frac{(1-k)^{2\alpha} + 1 - k^{2\alpha}}{2(1-k)^\alpha}\right) = f(\alpha, 1-k), \tag{3}$$

$$\theta_1^* + \theta_2^* = \arccos\left(\frac{1 - k^{2\alpha} - (1-k)^{2\alpha}}{2k^\alpha(1-k)^\alpha}\right) =: h(\alpha, k),$$

where we have defined the flow fraction $k := m_1/(m_1 + m_2)$ and the two functions $f$ and $h$, related via $h(\alpha, k) = f(\alpha, k) + f(\alpha, 1-k)$. If a BP exists that realizes the branching angles $\theta_i^*$, it can be constructed geometrically based on the *central angle property* (see App. A). It states that, given a circle through $a_1$ and $a_2$, the angle $\angle a_1 o a_2$ at the center $o$ is twice the angle enclosed with a point anywhere on the opposite circle arc, cf. Fig. 3c. In particular, let us construct the so-called *pivot circle* with central angle $\angle a_1 o a_2 = 2\theta_1^* + 2\theta_2^*$ and *pivot point* $p$ as in Fig. 3b. Applying the central angle property twice (once for $\theta_1^*$ and once for $\theta_2^*$), a BP located at the intersection of the lower circle arc and the connection line $\overline{a_0 p}$ realizes both angles $\theta_i^*$ and is therefore optimal.

However, given the pivot point and pivot circle, $\overline{a_0 p}$ may not intersect the lower circle arc, depending on the position of $a_0$. Accordingly, the lower half plane can be partitioned into a region for which the described construction yields an optimal Y-shaped branching and three other regions, see Fig. 5. For $a_0$ located in one of these regions, the optimal BP position coincides with one of the terminals, resulting in a V-shaped branching ($b^* = a_0$) or an L-shaped branching ($b^* \in \{a_1, a_2\}$), cf. Fig. 11 [2].

## 3.2 Geometric construction of BOT solutions for a given topology

Applying the geometric construction from above in a recursive manner, one can construct the ROS (see Def. 2.1) for larger BOT problems, as illustrated in Fig. 4. Given a full tree topology $T$, first, we determine all edge flows (see Sect. 2) and consequently the optimal branching angles. Then, a *root node* is chosen, arbitrarily (here $a_0$), and all other nodes are sorted based on the number of edges to $a_0$ (ignoring edge directions and resolving ties arbitrarily). Starting from the furthest nodes and working towards the root, two nodes are recursively summarized by a pivot point, constructed from the optimal branching angles, see Fig. 4a-b. Afterwards, in reversed order, the optimal BPs are placed iteratively, each as in the 1-to-2 case, see Fig. 4c-d. In this manner, the optimal branching angles are realized at every BP and the resulting solution is a ROS of $T$ by Lem. 2.1.

The choice of the root node induces a node ordering as described above. Given this ordering, consider any BP $b$ and denote its children by $a_1$ and $a_2$ and its parent node by $a_0$. The construction of the pivot point now requires the positions of $a_1$ and $a_2$ and the optimal branching angles enclosed by the children edges $(b, a_1)$ and $(b, a_2)$. However, the branching angles do not only depend on the absolute flows $m_1$ and $m_2$ of the respective edges but also on the flow directions. Given that both flows point towards $b$ or given that both flows point away from $b$, as in Fig. 3a, the branching is referred to as

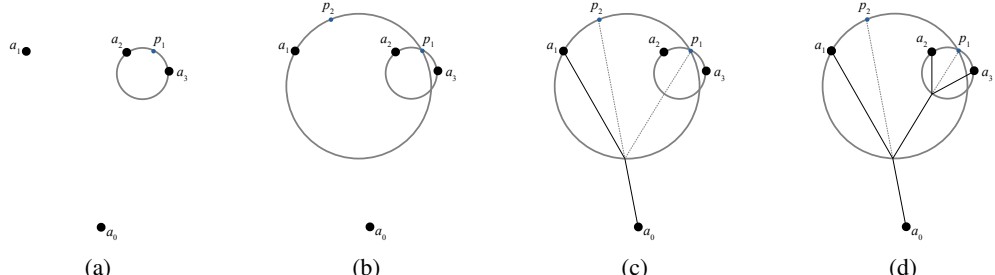

(a)      (b)      (c)      (d)

Figure 4: Recursive geometric construction of a relatively optimal solution, using one pivot point and pivot circle per branching point to collectively realize the optimal branching angles.

*symmetric* and the optimal branching angles of interest are given by $\theta_i^*$, cf. Eq. (3). Note that BOT problems and their solutions are fully symmetric under complete exchange of sinks and sources (up to reversal of all flow directions). On the contrary, in case of one flow pointing towards $b$ and one pointing away from $b$, referred to as *asymmetric branching*, the optimal branching angles $\vartheta_i^*$ enclosed by the children edges are calculated differently, see Fig. 3d. However, the branching angles $\vartheta_i^*$ can be related geometrically to the known $\theta_i^*$. Using the functions $f$ and $h$ from Eq. (3), we find that

$$
\begin{aligned}
\vartheta_1^* &= \pi - \theta_1^* - \theta_2^* = \pi - h\left(\alpha, k = \frac{m_2 - m_1}{m_2}\right) \\
\vartheta_2^* &= \theta_1^* = f\left(\alpha, k = \frac{m_2 - m_1}{m_2}\right).
\end{aligned}
\tag{4}
$$

After determining the two angles $\vartheta_1^*$ and $\vartheta_2^*$ from the flows $m_1$ and $m_2$, the BP construction based on the central angle property works analogously to the symmetric case. Crucially, this distinction of symmetric and asymmetric branching makes the recursive construction applicable also to problems with multiple sources, where asymmetric branchings may be unavoidable, consider e.g. Fig. 4d with $a_0$ and $a_3$ as sources and $a_1$ and $a_2$ as sinks (see App. G.1). Further, note that the known conditions for optimal V- and L-branching can be transferred to the asymmetric case simply by relabelling $a_0 \to a_2$, $a_1 \to a_0$ and $a_2 \to a_1$, cf. Fig. 3a and Fig. 3d. In terms of angular inequalities (derived in App. B), these conditions, for both branching types, are summarized in Table 1.

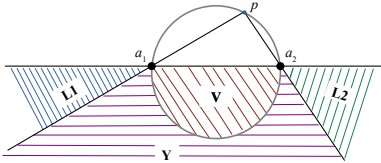

Figure 5: Regions of optimal Y-, V- and L-branching.

| | symmetric branching | asymmetric branching |
|---|---|---|
| V: | $\angle a_1 a_0 a_2 \geq \theta_1^* + \theta_2^*$ | L2: $\angle a_0 a_2 a_1 \geq \theta_1^* + \theta_2^*$ |
| L1: | $\angle a_2 a_1 a_0 \geq \pi - \theta_2^*$ | V: $\angle a_1 a_0 a_2 \geq \pi - \theta_2^*$ |
| L2: | $\angle a_0 a_2 a_1 \geq \pi - \theta_1^*$ | L1: $\angle a_2 a_1 a_0 \geq \pi - \theta_1^*$ |

Table 1: Relations between the L- and V-branching conditions.

In principle, given a full tree topology, the described method efficiently constructs the ROS in linear time. However, as already pointed out by Gilbert [9], the approach has some practical limitations, even after our generalization. Figure 3b shows how the pivot point is constructed only from the positions of two children $a_1$ and $a_2$ and the corresponding optimal branching angles. However, a priori there are two possible pivot point locations, one in the upper and one in the lower half plane with respect to $\overline{a_1 a_2}$. Hence, the construction relies on knowing in which half plane the third node $a_0$ lies. For larger trees, the topological parent $a_0$ may itself be a BP whose position is not yet determined. In the worst case, one would thus have to try all $2^{n-2}$ possible pivot point combinations to find the ROS. This pivot point degeneracy gets substantially worse in higher dimensions, making the recursive construction applicable only in $\mathbb{R}^2$. Secondly, the geometric construction only produces solutions which are non-degenerate, i.e., solutions without edge contractions. For now, the geometric construction is therefore primarily of theoretical interest; and indeed, it forms the basis of our following arguments. Note that both of the aforementioned problems could be solved elegantly in the special case of $\alpha = 0$ [12, 14].

# 4 Properties of optimal BOT topologies

Let us now consider topological modifications in order to improve the transportation cost of a BOT solution. In particular, we intend to show that a topology $T$ can be improved if its ROS contains coupled BPs. Let us start by considering a general BOT solution which contains a coupled 4-BP, i.e., a coupled BP with four effective neighbors, as in Fig. 2b. Lemma 2.1 states that a solution is not globally optimal if any subsolution is not globally optimal. It will therefore suffice to study the coupled BP as an isolated subproblem.

## 4.1 Non-optimality of coupled branching points

Given two sources and two sinks, there are two possible configurations in which the terminals can be arranged, cf. Fig. 6a,b. First we address the case in which the two sources are at opposite corners of the terminal quadrilateral, as in Fig. 6a. Based on Lem. 2.1, a necessary condition for the existence of a globally optimal 4-BP is that all four V-branchings between neighboring terminals are optimal. This puts a lower bound on each of the angles $\gamma_i$, see Tab. 1. The general idea, also regarding the other 4-branching scenarios, is to show that the angular sum of these lower bounds already exceeds $2\pi$. This will immediately imply that not all V-branchings can be optimal simultaneously and thus a coupled 4-BP cannot be globally optimal. Given a 4-BP as in Fig. 6a, all V-branchings are asymmetric (i.e. neighboring flows point in opposite directions). Hence, all four lower bounds (in Tab. 1) are of the form $\gamma_i \geq \pi - \theta_2^* = \pi - f(\alpha, 1 - k)$ and indeed $\pi - f(\alpha, 1 - k) > \pi/2$, see Lem. D.1, so that their sum exceeds $2\pi$.

Next, let us consider the scenario in Fig. 6b with two sources at neighboring corners. WLOG, we use the normalization $m_1 + m_2 = 1 = m_3 + m_4$ and assume that $m_1 > m_3$ and $m_2 < m_4$. In this case, the four conditions for optimal V-branching in Tab. 1 read:

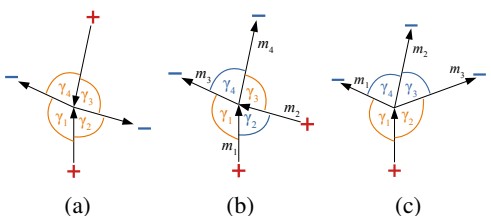

(a)    (b)    (c)

Figure 6: Different scenarios of coupled 4-BPs with symmetric branching angles in blue, asymmetric ones in orange.

$$\gamma_1 \geq \pi - f\left(\alpha, 1 - \frac{m_1 - m_3}{m_1}\right) = \pi - f\left(\alpha, \frac{m_3}{m_1}\right),$$

$$\gamma_2 \geq h\left(\alpha, \frac{m_1}{m_1 + m_2}\right) = h(\alpha, m_1),$$

$$\gamma_3 \geq \pi - f\left(\alpha, 1 - \frac{m_4 - m_2}{m_4}\right) = \pi - f\left(\alpha, \frac{1 - m_1}{1 - m_3}\right),$$

$$\gamma_4 \geq h\left(\alpha, \frac{m_3}{m_3 + m_4}\right) = h(\alpha, m_3),$$

where the expressions (3) were plugged into the V-branching conditions in Tab. 1 for symmetric and asymmetric branching respectively, as indicated by the colors in Fig. 6. Let us show that in fact for all combinations of $\alpha$, $m_1$ and $m_3$ the sum of the lower bounds already exceeds $2\pi$. Indeed, summing the lower bounds and subtracting $2\pi$ yields

$$\underbrace{h(m_1)}_{= h(1 - m_1)} + h(m_3) - \underbrace{f\left(\frac{m_3}{m_1}\right)}_{> m_3} - f\left(\underbrace{\frac{1 - m_1}{1 - m_3}}_{> 1 - m_1}\right) > h(1 - m_1) + h(m_3) - f(m_3) - f(1 - m_1)$$

$$= f(m_1) + f(1 - m_3) > 0,$$

using $h(\alpha, k) = f(\alpha, k) + f(\alpha, 1 - k)$ and the fact that $f(\alpha, k)$ is strictly decreasing with respect to $k$, see Lem. D.1. To summarize, we have arrived at the following lemma:

**Lemma 4.1.** *A coupled 4-BP not coincident with a terminal connecting two sources and two sinks is never globally optimal.*

Exactly the same logic applies for a coupled 4-BP connecting one source and three sinks (or equivalently 3 sources and 1 sink), as in Fig. 6c. WLOG, in the following, we normalize the flows so that $m_1 + m_2 + m_3 = 1$. We then determine the necessary conditions under which all V-branchings are optimal. We again intend to show that such a 4-BP can never be globally optimal by showing that for any combination of $\alpha$ and $m_i$ the sum of the lower bounds exceeds $2\pi$. This is equivalent to proving the following inequality (see App. E.1.1):

$$h\left(\frac{m_1}{m_1 + m_2}\right) - f(m_1) + h\left(\frac{m_3}{m_3 + m_2}\right) - f(m_3) > 0.$$

Assuming a globally optimal 4-BP existed, one could continuously displace a terminal in a way such that for the resulting BOT problem a coupled 4-BP is still globally optimal. Choosing different such displacements four additional inequalities can be derived (see App. E.1.2):

**Proposition 4.2.** *Given a BOT problem with one source and three sinks, with demands $m_1, m_2, m_3$ as in Fig. 6c, a coupled 4-BP away from the terminals cannot be globally optimal if at least one of the following inequalities holds true:*

$$\Gamma = h\Big(\frac{m_1}{m_1 + m_2}\Big) - f(m_1) + h\Big(\frac{m_3}{m_3 + m_2}\Big) - f(m_3) > 0,$$

$$\Gamma_{1,*} = f(1 - m_*) + f\Big(1 - \frac{m_2}{1 - m_*}\Big) - f(1 - m_* - m_2) > 0,$$

$$\Gamma_{2,*} = h\Big(\frac{m_*}{m_* + m_2}\Big) + f\Big(\frac{m_2}{1 - m_*}\Big) - h(m_*) > 0$$

*where $* = 1, 3$. Note that $\Gamma = \Gamma_{1,1} + \Gamma_{2,1} = \Gamma_{1,3} + \Gamma_{2,3}$.*

In App. E.1.3, we prove the inequalities analytically for a large subset of the parameter space. For the remainder we present a numerical argument (see App. E.1.4). In addition, we show by induction how, given that coupled 4-BPs are never globally optimal, one can further rule out coupled $n$-BPs (with $n$ effective neighbors) for all $n > 4$.

**Theorem 4.3.** *Given a BOT problem in the Euclidean plane and assuming that coupled 4-BPs are never globally optimal, in a globally optimal BOT solution each branching point not coincident with a terminal must have degree three.*

## 5 Generalization of BOT to Riemannian manifolds

In this section, we extend the BOT problem together with many of the previous results to two-dimensional Riemannian manifolds $\mathcal{M}$ embedded into $\mathbb{R}^3$ [18]. This includes the sphere as important special case, particularly relevant for global transportation networks. In the generalized BOT cost function (5) we replace the Euclidean metric by the geodesic distance $d : \mathcal{M} \times \mathcal{M} \to \mathbb{R}^+$, i.e.

$$\mathcal{C}_M = \sum_{(i,j) \in E} m_{ij}^\alpha \, d(x_i, x_j). \tag{5}$$

As we assume the manifold to be embedded, the length of a geodesics can be measured in $\mathbb{R}^3$. First, we generalize the non-optimality of cyclic solutions. The corresponding proof in [2] readily applies also to two- and higher-dimensional manifolds. As before, solving a BOT problem on a curved surface can thus be separated into the combinatorial topology optimization and the continuous optimization of the BP configuration.

### 5.1 Linear approximation of BOT solutions on manifolds

Intuitively speaking, a two-dimensional Riemannian manifold locally looks like the Euclidean plane. If we zoom in on a sufficiently small region, geodesics again resemble straight lines and the geodesic distance approaches the Euclidean one. This can be used to show that the branching angles which were optimal for Y-branchings in the Euclidean plane are also optimal on Riemannian manifolds. Below, we summarize the main steps of the proof. All details can be found in App. F.

Given a Y-branching on a manifold, we measure the angles between the three geodesics in the tangent space $T_b\mathcal{M}$ at the BP $b$. We now zoom in on a small neighborhood $U$ around $b$ and consider only the subsolution in $\mathcal{M} \cap U$. The terminals of the corresponding subproblem are projected orthogonally onto the tangent space, more specifically onto a small disk of radius $r$, denoted by $D(r)$, see Fig. 7. Let us denote the cost of the subsolution on the manifold by $\mathcal{C}_M(b)$ and the cost of the corresponding subproblem in the flat disk by $\mathcal{C}(b)$. Now, assuming that the angles between the geodesics deviate from the optimal branching angles, the same holds true for the projected subsolution. Consequently, there exists an alternative BP $b^*$ in the disk with cheaper cost $\mathcal{C}(b^*)$. Note that the radius of this disk becomes smaller the smaller we choose the region $\mathcal{M} \cap U$ of the subproblem.

Crucially, the cost difference between a subsolution on the manifold and its projection onto the plane tends to zero quadratically in the limit of $r \to 0$. The intuitive reason for this is that the tangent space $T_b\mathcal{M}$ locally approximates the manifold to linear order. On the contrary, the costs $\mathcal{C}(b)$ and $\mathcal{C}(b^*)$ in the disk scale linearly in $r$ and so does the cost improvement $\mathcal{C}(b) - \mathcal{C}(b^*) = M\,r$, for some fixed $M > 0$. To conclude the proof, one projects $b^*$ onto the manifold and evaluates the cost difference of the two subsolutions there. The difference is of the form $M\,r + O(r^2)$, with second order differences due to the projection from $D(r)$ to the manifold. Consequently, a finite radius $r > 0$ must exist for which the cost difference is truly positive. A BOT solution on the manifold for which the Y-branching angles deviate from the optimal branching angles can thus be improved and is not relatively optimal. The logic of the proof outlined here can easily be extended to the V- and L-branching conditions as well as our results regarding the non-optimality of coupled

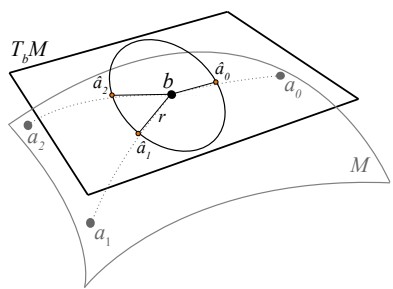

Figure 7: On the flat disk $D(r) \subset T_b\mathcal{M}$, the BP $b$ does *not* solve the problem with terminals $\hat{a}_i$ optimally if the angles between the dotted geodesics are not optimal.

BPs. Again, improving the BOT solution locally in the tangent plane (w.r.t. its geometry or topology) and projecting back to the manifold results in an improved solution on the manifold (see App. F.2).

**Theorem 5.1.** *Consider the solution to a generalized BOT problem on a two-dimensional Riemannian manifold embedded into $\mathbb{R}^3$. For the solution to be relatively optimal, it is a necessary condition that each BP satisfies the optimal angle conditions for Y-, V- and L-branching, which apply for BOT in the Euclidean plane. For it to be globally optimal, assuming that coupled 4-BPs are not optimal in the plane, it is a necessary condition that BPs not coincident with a terminal have degree three.*

Though there is no readily available algorithm to solve BOT on embedded surfaces, we discuss some possible approaches in App. F.3.

# 6 Heuristics and numerical optimization

In this section, we present a simple but effective algorithm for the geometry optimization, followed by a compelling heuristic for the topology optimization. As pointed out earlier, the difficulty of solving a BOT problem stems from the super-exponentially growing number of possible full tree topologies. Obtaining an exact solution by brute-force is almost always computationally infeasible and hence fast heuristic solvers are needed. For BOT problems with a single source, a branch-and-bound method is applicable [31], enabling exact solutions for up to 16 nodes. However, this method does not generalize directly to the case of multiple sources. While some literature exists on heuristics for BOT problems with a single source [29], we are not aware of heuristics for multiple sources, except [24]. The authors of [24] present a simulated annealing based optimization strategy for BOT, based on hand-crafted geometrical and topological modifications, which may require user supervision. Furthermore, continuous approaches to solve BOT exist which do not rely on a subdivision into geometry and topology optimization. The authors of [22] phrase BOT as a limit of functional minimization problems. Since their algorithm discretizes the plane and the BOT cost function, their output is however not sparse but a discretized function.

## 6.1 Numerical branching point optimization for a given topology

Brute-force and heuristic BOT solvers alike typically rely on the geometry optimization of many different topologies. A fast and reliable BP optimization routine is therefore essential, as it determines the computational bottleneck of these algorithms. For a given tree topology $T$, all edge flows $m_{ij}$ are known (see Sect. 2). The objective is thus to minimize the following convex cost function:

$$\mathcal{C}(\{x_i\}) = \sum_{(i,j)\,\in\,T} m_{ij}^{\alpha} \, \|x_i - x_j\|_2 \,, \tag{6}$$

where, for $1 \leq i \leq n$, the $x_i$ hold the fixed coordinates of the terminals and, for $n+1 \leq i \leq n+m$, the variable BP positions. Since the cost function is not everywhere differentiable, we suggest the following generalization of Smith's algorithm developed for geometry optimization in the ESTP [26].

It is an effective algorithm specifically for minimizing the sum of Euclidean norms in two- and higher-dimensional Euclidean space. Unlike the geometric construction in Section 3.2, it is applicable to all (not necessarily full) tree topologies.

Starting from a non-optimal, non-degenerate BP configuration, e.g. from a random initialization, the gradient with respect to each BP position $x_i$ is set to zero for $n + 1 \leq i \leq n + m$, resulting in the following non-linear system of $m$ equations:

$$x_i = \sum_{j\,:\,(i,j)\in T} m_{ij}^\alpha \frac{x_j}{|x_i - x_j|} \Bigg/ \sum_{j\,:\,(i,j)\in T} \frac{m_{ij}^\alpha}{|x_i - x_j|}.$$

This system can be solved approximately, by iteratively solving the following *linearized* system

$$x_i^{(k+1)} = \sum_{j\,:\,(i,j)\in T} m_{ij}^\alpha \frac{x_j^{(k+1)}}{|x_i^{(k)} - x_j^{(k)}|} \Bigg/ \sum_{j\,:\,(i,j)\in T} \frac{m_{ij}^\alpha}{|x_i^{(k)} - x_j^{(k)}|}, \quad \text{for } n + 1 \leq i \leq n + m. \quad (7)$$

Note that $x_i^{(k)} = x_i$ is fixed for $1 \leq i \leq n$. For each iteration, the solution can be found in linear time, again by "elimination on leaves of a tree", similar to determining all edge flows from the flow constraints. The algorithm is easily parallelized over $d$ spatial dimensions of a BOT problem so that a single iteration is of order $O(nd)$. In essence, this is an iteratively reweighted least squares (IRLS) approach [4]. The connection is made explicit in App. G.2. Details on the proof of convergence, the empirical runtime of the algorithm and suitable convergence criteria can be found in App. G.2 and in [26]. The arguments in [26] readily apply to our generalization. Besides our method, other techniques may be used for the geometry optimization, for instance the interior point method presented in [30].

## 6.2 A greedy randomized algorithm for the topology optimization

Our heuristic for the optimization of the BOT topology is inspired by the idea of simulated annealing [16], which has been applied in different variants to combinatorial problems such as the Traveling Salesman Problem [19] or the ESTP [10]. In our heuristic, the BOT topology is iteratively modified by randomly deleting an edge and replacing it with a new one. At each step, the new solution is accepted according to a criterion, which typically depends on the cost difference between the solutions and a user-chosen hyperparameter, the temperature, used to mimic a physical cooling process. However, because in practice it works already sufficiently well (see Fig. 8), we refrained from designing an elaborate cooling scheme. Instead, we apply the heuristic most greedily, i.e., in the zero-temperature limit, where a new state is accepted *only* if it decreases the cost.

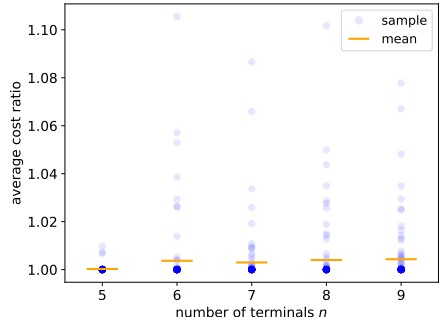

Figure 8: Cost ratios of our greedy heuristic and brute-force solutions (the closer to 1 the better) for different number of terminals $n$. For each $n$, we uniformly sampled 100 different BOT problems. Most runs ended up close to the global optimum (see dark blue assembly near 1.0).

Starting from an initial tree topology $T$, e.g., the minimum spanning tree (mST) or the OT solution[2], we uniformly sample an edge $\hat{e} \in E$ and remove it from $T$. Let the incident node of $\hat{e}$ which ended up in the smaller connected component be $\ell$. Then, one calculates the distance $d(e, \ell)$ between $\ell$ and every edge $e = (i, j)$ in the larger component and samples one of these edges with probability $p(e) \propto \exp(-d(e, \ell)^2 / d_{min}^2)$, where $d_{min}$ is the distance to the closest considered edge. The node $\ell$ is then connected to the sampled edge via a new BP to produce a new tree topology. For this topology, we optimize the geometry (as described in Sect. 6.1) and compare costs with the previous solution. If the new state is rejected, start the next iteration by sampling $\hat{e}$ *without replacement* until either a move is accepted and all above steps are repeated; or until no accepted move is found, upon which the search terminates.

---

[2]In particular in the regime $\alpha \approx 1$, our BOT solver benefits from existing efficient OT solvers by using their solution as initial guess.

Experiments for small BOT problems suggest that even in the greedy zero-temperature limit the algorithm often finds the globally optimal solution, after comparatively few iterations. For this, the greedy heuristic (using the mST as initialization) was compared against exact solutions with up to nine terminals, obtained by brute-force. For each $n$, 100 BOT problems were sampled uniformly with respect to $\alpha$, the terminal positions and demands and supplies, cf. Alg. 2. The ratios of the heuristic's cost divided by the cost of the exact solution are plotted in Fig. 8. On average the heuristic solution is less than 0.5% worse than the brute-force solution. This is impressive, considering the fact that for $n = 9$ the brute-force solver requires over $10^5$ BP optimizations, whereas the simulated annealing heuristic on average required $29 \pm 10$ iterations to converge. Additional experiments (also for larger BOT problems) suggest that the number of BP optimizations until convergence scales better than $O(n^2)$, see App. G.3. Further, the cost ratios in Fig. 8 stay roughly constant as $n$ increases. Additional experiments for BOT in higher dimensions (see Fig. 28) indicate that the average quality of the heuristic solution decreases only very slightly with $n$. Unfortunately, one can only speculate how this trend extends to larger BOT problems, where brute-force solutions are no longer feasible. Figure 1 shows heuristic solutions of a larger example problem for different values of $\alpha$. In particular, we find that the greedy heuristic is very effective at removing higher-degree branchings and undesirable edge crossings.

## 7 Generalization to higher-dimensional BOT

Optimal BOT solutions are acyclic also in $\mathbb{R}^d$ [2]. Thus, for a given topology, the edge flows are known, the optimal substructure property of Lemma 2.1 generalizes and the convex geometry optimization can be separated from the combinatorial topology optimization. Though, the optimal angle conditions for Y-, V- and L-branching (see Sect. 3) hold also in $\mathbb{R}^d$, the results on the degree limitation do not generalize, as the arguments rely on the fact that the angles between edges meeting at a higher-degree branching point sum up to $2\pi$ (cf. Sect. 4.1). The numerical geometry optimization as well as the greedy algorithm for the topology optimization presented in Sect. 6 are readily applicable to BOT problems in $\mathbb{R}^d$ (see also App. G.2 and App. G.3).

## 8 Conclusions

We have studied branched optimal transport in $\mathbb{R}^2$ from a theoretical and practical perspective. First, we have tackled the geometric optimization of BOT solutions, given a tree topology. We generalized the existing exact method presented in [2, 9] to the case of multiple sources. Based on theory developed in the process of this generalization, we formulated a catalog of necessary and sufficient conditions for optimal BOT solutions and argued that $n$-degree branching points for $n > 3$ are never optimal. Moreover, we showed that these conditions also apply for BOT on two-dimensional manifolds. Lastly, we presented a greedy randomized algorithm, which optimizes the tree topology, combined with an efficient numerical branching point optimization method. We compared our algorithm to the optimal solution for small examples, obtaining compelling results.

BOT provides a unifying framework for optimal transport and the Euclidean Steiner tree problem and is itself of great theoretical and practical interest. The emergent branching in BOT can be used to simulate and study the myriad of efficient transportation systems which exhibit subadditive costs. Moreover, BOT combines both combinatorial and convex optimization and could be an inspiring problem to be solved by machine learning techniques. The number of optimality criteria derived in this paper can guide further research in this area and the presented approximate solvers may serve as competitive baseline for new ML-based approaches.

## Acknowledgments and Disclosure of Funding

We would like to thank Jarosław Piersa for sharing his code with us for a comparison to his work. Further, we thank Edouard Oudet for helpful hints on the comparison to his related work and Fabian Egersdoerfer for his improved C++ implementation of the geometry optimization.

This work is supported by Deutsche Forschungsgemeinschaft (DFG) under Germany's Excellence Strategy EXC-2181/1 - 390900948 (the Heidelberg STRUCTURES Excellence Cluster), by Informatics for Life and by SIMPLAIX funded by the Klaus Tschira Foundation.

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
