## Appendix overview

The appendix is subdivided into the following seven topics:

**A Central angle property**: Quick proof of the central angle property, used in Sect. 3.2 in the recursive construction of relatively optimal solutions based on the optimal branching angles.

**B Optimal L- and V-shaped branching**: Derivation of the conditions listed in Tab. 1 under which V- or L-branching are optimal.

**C Optimal substructure of BOT solutions (Proof of Lem. 2.1)**: Formal proof of Lemma 2.1 on the optimal substructure of BOT solutions.

**D Properties of the functions $f$ and $h$ describing the optimal branching angles**: Collection of small lemmas on the monotonicity and other properties of the analytical expression for the branching angles (cf. Eq. (3)).

**E Non-optimality of higher-degree branchings**: Technical proofs and numerical scheme to show the non-optimality of higher-degree branchings discussed in Sect. 4.1.

**F BOT on two-dimensional Riemannian manifolds**: Formal proof of Theorem 5.1, which generalizes the optimal branching conditions and other properties from the Euclidean plane to embedded surfaces. A sketch of the proof can be found in Sect. 5.1.

**G Algorithms**: Additional details and experiments for the different algorithms presented in the main paper. Section G.2 focuses on the numerical geometry optimization and Sect. G.3 on the greedy algorithm for the topology optimization. Section G.1 holds a few examples of the recursive geometric construction of relatively optimal solutions for BOT problems with multiple sources.

## A Central angle property

In this section, we present a geometric proof of the central angle property used in the geometric construction of relatively optimal solutions for a given full tree topology (see Sect. 3.2). It states that for a circle, as in Fig. 9a, the central angle $\angle a_1 o a_2$ is twice the angle $\angle a_1 q a_2 = \theta$ for all $q$ on the lower circle arc.

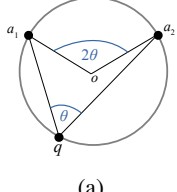 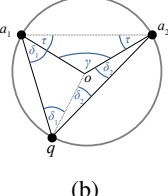

(a)                    (b)

Figure 9: **(a)** Illustration of the central angle theorem. **(b)** Isosceles triangles used in the geometric proof.

Let us start by constructing the three isosceles triangles $\triangle oqa_1$, $\triangle oqa_2$ and $\triangle oa_1 a_2$ with angles as denoted in Fig. 9b. Now consider the angular sums in the following triangles:

$$\triangle qa_1a_2 :\ 2\delta_1 + 2\delta_2 + 2\tau = 180^\circ \ \text{ and } \ \triangle oa_1a_2 : \gamma + 2\tau = 180^\circ \ .$$

Subtracting the two equations immediately reveals that $\gamma = 2\delta_1 + 2\delta_2 = 2\theta$ and the proof is complete.

## B Optimal L- and V-shaped branching

Below, we formally derive the conditions listed in Tab. 1 under which V- or L-branching provide the optimal solution to a BOT problem with one source and two sinks. The proof is inspired by the approach in [21], where subdifferentials are applied to the related Fermat-Torricelli problem.

**Definition B.1** (Subgradient and subdifferential). A vector $v \in \mathbb{R}^n$ is called a *subgradient* of a convex scalar function $g : \mathbb{R}^n \to \mathbb{R}$ at a specific point $y$ if for all $x \in \mathbb{R}^n$ it satisfies

$$g(x) \geq g(y) + \langle v, x - y \rangle \tag{8}$$

The set of all subgradients of the function $g$ at a given point $y$ is called the *subdifferential* of $g$ at $y$ and is denoted by $\partial g(y)$.

From a geometric point of view, the subdifferential of $g$ at $y$ is the set of gradients of all straight lines which cross $g(y)$ and lie below the image of $g$. The subdifferential rule of Fermat follows immediately from the definition and states that $g$ achieves an absolute minimum at $y$ if and only if $0 \in \partial g(y)$. Now, for $g(b) = c \cdot \|b - a\|_2$ with $c \in \mathbb{R}$ and $a \in \mathbb{R}^n$ the subdifferential is given by

$$\partial g(b) = \begin{cases} \mathbb{B}_c \,, & \text{for } b = a \\ \left\{ c \frac{b-a}{\|b-a\|_2} \right\}, & \text{elsewhere,} \end{cases}$$

with $B_r = \{v \in \mathbb{R}^n : \|v\|_2 \leq r\}$, the ball of radius $r$. Furthermore, it can be easily shown that for $g(x) = g_1(x) + g_2(x)$, one has $\partial g(y) = \partial g_1(y) + \nabla g_2(y)$, given that both $g_1$ and $g_2$ are convex functions and $g_2$ is differentiable.[3] Using this, we calculate the subdifferentials of the cost function of the 1-to-2 branching in Eq. (2). The subdifferentials at $b = a_i$ are of the form

$$\partial \mathcal{C}(a_i) = \mathbb{B}_{m_i^\alpha} + \sum_{j \neq i} m_j^\alpha \frac{a_i - a_j}{\|a_i - a_j\|} \, .$$

Based on the rule of Fermat, the cost function achieves an absolute minimum at $b = a_i$ if and only if

$$\left\| \sum_{j \neq i} m_j^\alpha \frac{a_i - a_j}{\|a_i - a_j\|} \right\| \leq m_i^\alpha \, . \tag{9}$$

**V-branching.**  We square condition (9) and evaluate it for $i = 0$ in order to determine under which condition a V-shaped branching with $b^* = a_0$ is optimal:

$$\left\| m_1^\alpha \frac{a_0 - a_1}{\|a_0 - a_1\|} + m_2^\alpha \frac{a_0 - a_2}{\|a_0 - a_2\|} \right\|^2 = m_1^{2\alpha} + m_2^{2\alpha} + 2 m_1^\alpha m_2^\alpha \cos(\psi) \leq m_0^{2\alpha} = (m_1 + m_2)^{2\alpha}$$

where $\psi$ denotes the angle of the terminal triangle at $a_0$, i.e., $\psi = \angle a_1 a_0 a_2$. The condition in terms of $\psi$ reads

$$\psi \geq \arccos \left( \frac{1 - k^{2\alpha} - (1-k)^{2\alpha}}{2 k^\alpha (1-k)^\alpha} \right) = h(\alpha, k) = \theta_1^* + \theta_2^* \, , \tag{10}$$

where we have used the flow fraction $k = m_1/(m_1 + m_2)$. We immediately recognize the expression for the optimal branching angle $\theta_1^* + \theta_2^*$, cf. Eq. (3).

We already know that $\psi = \theta_1^* + \theta_2^*$ on the lower circle arc of the pivot circle by construction. And indeed one can easily check that $\psi > \theta_1^* + \theta_2^*$ if an only if the source $a_0$ lies inside the lower half of the pivot circle, as in Fig. 10. For that, we construct a line through $a_1$ and $a_0$ and the intersection of $\overline{a_1 a_0}$ with the lower pivot circle we denote by $q$. By construction of the pivot circle, $\angle a_1 q a_2 = \theta_1^* + \theta_2^*$. Using the angular sum in the triangle $\triangle a_0 q a_2$, one immediately obtains:

$$180° = \delta + (180° - \psi) + (\theta_1^* + \theta_2^*) \; \to \; \psi - (\theta_1^* + \theta_2^*) = \delta > 0 \, .$$

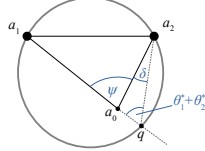

Figure 10: Sketch to show that $\psi > \theta_1^* + \theta_2^*$.

Hence, for $a_0$ located inside the lower half of the pivot circle indeed $\psi > \theta_1^* + \theta_2^*$. A similar argument can show that for any point outside the pivot circle the optimal V-branching condition is not fulfilled.

---

[3]The sum of a vector and a set of vectors, as in $\partial g_1(y) + \nabla g_2(y)$, is known as the Minkowski sum.

**L-branching.** Analogous to the above steps, one obtains conditions for optimal L1- and L2-branching, where $b^* = a_1$ and $b^* = a_2$ respectively. Again squaring the general condition (9), now for $i = 1, 2$, one eventually finds that

$$\varphi \geq \pi - f(\alpha, 1-k) = \pi - \theta_2^* , \qquad (11)$$
$$\varrho \geq \pi - f(\alpha, k) = \pi - \theta_1^* , \qquad (12)$$

where the angle $\varphi$ and $\varrho$ denote the angles of the terminal triangle located at $a_1$ and $a_2$, i.e. $\varphi = \angle a_2 a_1 a_0$ and $\varrho = \angle a_0 a_2 a_1$ (see also Fig. 11c,d). Let us now demonstrate that these conditions are indeed fulfilled if and only if the source is located in the L1- and L2-region, as marked also in Fig. 5. The pivot point $p$ is constructed such that $\angle a_1 o p = 2\theta_1^*$ and $\angle p o a_2 = 2\theta_2^*$, cf. Fig. 3b. Besides that, by construction, we have that $\angle p a_1 a_2 = \theta_2^*$ and $\angle a_1 a_2 p = \theta_1^*$ (as shown in Fig. 11c,d). Then, looking at Fig. 11c, it is evident that indeed for any source $a_0$ inside the L1-branching sector the condition $\varphi = \angle a_2 a_1 a_0 \geq \pi - \theta_2^*$ holds. The respective condition (12) for L2-branching holds true exactly inside the highlighted L2-region.

**Definition B.2** (Transient and strict V- and L-branchings). A V- or L-branching for which the inequality conditions (10)-(12) hold as equality is referred to as *transient* V- or L-branching. The reason for this is that, in such a case, one of the terminal positions may be perturbed infinitesimally, so that the condition is no longer fulfilled and the optimal solution transitions to a Y-shaped branching. On the contrary, if an L- or V-branching condition is fulfilled as strict inequality, we call the L- or V-branching *strict*.

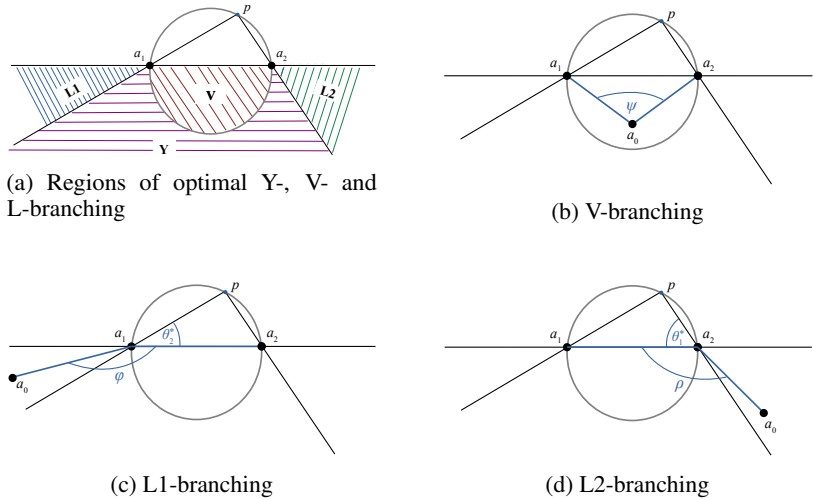

(a) Regions of optimal Y-, V- and L-branching

(b) V-branching

(c) L1-branching

(d) L2-branching

Figure 11: After constructing the pivot point the position of the source $a_0$ determines whether a Y-,V- or L-shaped branching is optimal.

## B.1 Relation of V- and L-conditions between symmetric and asymmetric branchings

Let us briefly show that the corner cases of Y-, V- and L-branching work analogously for both flow scenarios (described in Fig. 3a,d). The conditions under which L- and V-branching are optimal in the asymmetric branching case could be again determined straightforwardly by plugging into the subdifferential condition (9), as before. However, looking at Fig. 3a,d, we notice that the well-known symmetric branching case can be transformed into the asymmetric case by relabeling $a_0 \rightarrow a_2$, $a_1 \rightarrow a_0$ and $a_2 \rightarrow a_1$. This relation provides a direct correspondence of the L- and V-branching conditions. The conditions for the corner case are transferred according to Table 2. Note that the L- and V-branching conditions for both cases are of the exact same form, only that for the asymmetric case the stationary branching angles are $\vartheta_i^*$ instead of $\theta_i^*$. Moreover, for fixed children positions $a_1$ and $a_2$, the position of the source again distinguishes between optimal Y-, V- and L-branching. The partitioning of the lower half plane into the respective regions is completely analogous to Fig. 11a.

| symmetric branching | | asymmetric branching | |
|---|---|---|---|
| V: $\angle a_1 a_0 a_2 \geq \theta_1^* + \theta_2^*$ | | L2: $\angle a_0 a_2 a_1 \geq \theta_1^* + \theta_2^* = \pi - \vartheta_1^*$ | |
| L1: $\angle a_2 a_1 a_0 \geq \pi - \theta_2^*$ | | V: $\angle a_1 a_0 a_2 \geq \pi - \theta_2^* = \vartheta_1^* + \vartheta_2^*$ | |
| L2: $\angle a_0 a_2 a_1 \geq \pi - \theta_1^*$ | | L1: $\angle a_2 a_1 a_0 \geq \pi - \theta_1^* = \pi - \vartheta_2^*$ | |

Table 2: Relations between the L- and V-branching conditions (see Fig. 11) for symmetric and asymmetric branchings.

## C  Optimal substructure of BOT solutions (Proof of Lem. 2.1)

In this section, we provide the formal proof to Lemma 2.1, repeated below for completeness:

**Lemma C.1.** *(a) For a given tree topology, a BOT solutions is a relatively optimal if and only if every (coupled) BP connects its (effective) neighbors at minimal cost. (b) In a globally optimal solution, every subsolution restricted to a connected subset of nodes solves its respective subproblem globally optimally.*

Let us start with the following definition, which divides a BOT problem and its possible solutions into subproblems and corresponding subsolutions.

**Definition C.1** (Subproblems and subsolutions). A given BOT solution may be split into two subsolutions, by choosing a number of edges $\{e_i\}$ and cutting them at points $\{x_i\}$, so that the topology is split into two connected components. This procedure induces two subproblems and two subtopologies. Each *subproblem* consist of the terminals contained in the respective component plus additional terminals at the positions $x_i$. The demands or supplies of the additional terminals at $x_i$ are equal to the amount of flow through the corresponding edge $e_i$ that was cut. The terminal becomes a sink in one subproblem and a source in the other according to the direction of flow through $e_i$. The two *subtopologies* are given by the induced subgraph on all terminals contained in one component. The *subsolutions* to the created subproblems are given by the subtopologies and the BP configurations of the respective subsets of branching points contained in each subproblem. Note that using a number of such splits a given solution may be divided into several subsolutions, each solving their respective subproblem. An illustrative example can be found in Fig 12.

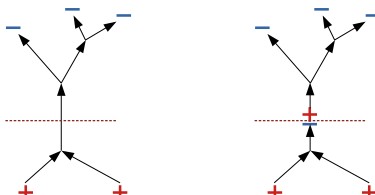

Figure 12: A BOT solution (left) is split into two subsolutions (right) by cutting an edge as indicated by the dashed line.

**Proof of the optimal substructure as necessary condition for optimality.** The optimal substructure as necessary optimality condition follows immediately from the independence of the subproblems. Given that a solution is minimal in cost (in the relative or global sense), each subproblems itself must be minimal in that sense. Otherwise the cost of this subproblem could be decreased by (a) improving the BP configuration or (b) the topology. In that case, also the full solution could be improved in cost and could thus not be optimal.

**Proof of the optimal substructure as sufficient condition for optimality.** The optimal substructure as sufficient optimality condition stems from the fact that, given a tree topology, the BOT cost function is convex with respect to the BP positions. As a first step, let us prove the following lemmata:

**Lemma C.2.** *Let $b$ be a branching point with neighbors $\{a_i\}$. Let the optimal position of $b$ be at one of its neighbors with the condition (9) fulfilled as equality, i.e. the branching is transient. Then, for*

*every $\delta > 0$ there exists a BP location $b_\delta \in \mathbb{R}^2$ away from all neighbors with $\|b - b_\delta\|_2 < \delta$ and a continuous function $f : \mathbb{R}^+ \to \mathbb{R}$ with $\lim_{x \to 0} f(x) = 0$, such that the gradient $\|\nabla_b \mathcal{C}(b_\delta)\|_2 < \epsilon$ for $0 < \epsilon < f(\delta)$.*

*Proof.* Let $\mathcal{C}(b)$ be the cost of the subproblem with terminals $a_i$. WLOG, the condition (9) is fulfilled as equality for terminal $a_0$, meaning that

$$\|v\| = m_0^\alpha \ \text{ with } \ v := \sum_{i \neq 0} m_i^\alpha \frac{a_0 - a_i}{\|a_0 - a_i\|} \ . \tag{13}$$

The optimal branching point is hence located at $a_0$. Consider the alternative branching point position $b_\delta = a_0 - \delta v$ with sufficiently small $\delta > 0$, such that $b_\delta$ is located away from all terminals. The gradient with respect to the branching point position is well defined at $b_\delta$ and reads:

$$\nabla_b \mathcal{C}(b_\delta) = -m_0^\alpha \frac{v}{\|v\|} + \sum_{i \neq 0} m_i^\alpha \frac{a_0 - a_i - \delta v}{\|a_0 - a_i - \delta v\|} \ . \tag{14}$$

Clearly, $\|\nabla_b \mathcal{C}(b_\delta)\|$ can be brought arbitrarily close to zero as $\delta \to 0$, since

$$\|\nabla_b \mathcal{C}(b_\delta)\|^2 = \underbrace{m_0^{2\alpha} + m_0^{2\alpha} - 2m_0^\alpha \left\langle v, \frac{v}{\|v\|} \right\rangle}_{= 0} + O(\delta) \ , \tag{15}$$

comprising the terms which go to zero as $\delta \to 0$ in $O(\delta)$. The function absorbed in $O(\delta)$ therefore provides the function $f$ and $b_\delta$ is the desired branching point with arbitrarily small gradient. The argument works for both coupled and uncoupled branching points. $\qquad \square$

**Lemma C.3.** *For a given full tree topology $T$, let $B = \{b_i\}$ be the ROS of a BOT problem with terminals $A = \{a_i\}$. Let $B$ contain Y-branchings and transient V- or L-branchings. Then, for every $\delta > 0$ there exists a non-degenerate BP configuration $B_\delta$ and a continuous function $f : \mathbb{R}^+ \to \mathbb{R}$ with $\lim_{x \to 0} f(x) = 0$, such that $\|B - B_\delta\|_2 < \delta$ and the gradient norm fulfills $\|\nabla_B \mathcal{C}(B_\delta)\|_2 < \epsilon$ for $0 < \epsilon < f(\delta)$.*

*Proof.* Let us split the solutions into subsolutions at branching points that exhibit optimal Y-branchings. They are kept fixed $\delta_i = 0$ and have zero gradients. The resulting subproblems can then be considered independently. It suffices to show that, for every individual subproblem, individual displacements of the BPs exist that yield a non-degenerate BP configuration with arbitrarily small gradients. If a subproblem consist of a single branching point, Lem. C.2 is directly applicable and the infinitesimal displacement is chosen as described there. If multiple branching points are coupled in a transient branching we proceed as follows: For the given topology, define a root node and apply the recursive geometric construction with pivot points and pivot circles, as illustrated in Fig. 4. In the construction, all pivot circles will meet at the position of the coupled branching point, which is the geometric equivalent of condition (9) being fulfilled as equality, cf. Fig. 16. Then, starting from the branching point furthest from the root node, take an infinitesimal step of size $\sim O(\delta)$ towards its corresponding pivot point, exactly as in Lem. C.2. Thereby, the optimal branching angles will almost be realized and consequently the gradient of the resulting uncoupled BP will be arbitrarily small (as shown explicitly in Lem. C.2). For the next BP in topological order (w.r.t. the chosen root node) repeat the procedure with a step size even smaller of size $\sim O(\delta^2)$, again the optimal branching angles are almost realized and the resulting gradient vanishes as $\delta \to 0$. Repeating, this procedure for every of the finitely many branching points with smaller and smaller step sizes produces a non-degenerate BP configuration with arbitrarily small gradient. $\qquad \square$

Based on Lem. C.3, we now prove that the optimality of each individual branching point is a sufficient condition for relatively optimal solutions. Similar to the proof above we do not explicitly distinguish between coupled and uncoupled branching points. Given a fixed tree topology, the BOT cost function $\mathcal{C}(\{b_i\})$ is a convex function of the branching point positions $\{b_i\}_{1 \le i \le m}$. Let us summarize all branching point coordinates in the vector $B \in \mathbb{R}^{2m}$ and denote the configuration in which every BP connects its neighbors at minimal cost by $B^* = \{b_i^*\}$. Due to the convexity it suffices to show that the BP configuration $\{b_i^*\}$ is a local minimum of the cost function. We may distinguish the following three cases:

(a) If the BP configuration $\{b_i^*\}$ is non-degenerate, the cost function is differentiable and for each BP one has $\nabla_{b_i} \mathcal{C}(b_i^*) = 0$ and thus also $\nabla_B \mathcal{C}(B^*) = 0$.

(b) Let the BP configuration additionally contain BPs which are strictly anchored at one of their neighbors (referred to as anchor), meaning that the inequality condition (9) holds strictly (see also Def. B.2). Denote the subset of these BPs by $\{\tilde{b}_j\}$. We intend to show that a neighborhood around $B^*$ exists such that $B^*$ is the minimal cost configuration in it. If the neighboring branching points of $\{\tilde{b}_j\}$ are moved away from $B^*$ only by a sufficiently small $\delta$, the optimal position for $\{\tilde{b}_j\}$ stays at their anchors, since the condition (9) will still hold. Thus, we can find a sufficiently small neighborhood $U$ around $B^*$, so that WLOG, the coordinates of $\{\tilde{b}_j\}$ are fixed to be equal to the coordinates of their respective anchors, as other configurations would be suboptimal. If a branching point is anchored at an external node, its position is fixed to the terminal position and the cost function restricted to $U$ no longer depends on them. So, by choosing $U$ sufficiently small, the cost function depends only on BPs away from terminals. Summarize the remaining free branching points in the vector $B_f$ for which the gradient at $B_f^* \subset B^*$ is zero by assumption, i.e. $\nabla_{B_f} \mathcal{C}(B_f^*) = 0$. Thus, $B^*$ is a local minimum in $U$ and due to the convexity an absolute minimum.

(c) Let the BP configuration additionally contain transient branchings, i.e. branching points for which the condition (9) holds as equality. Denote them by $\{\hat{b}_k\}$. Again, the sufficiently small neighborhood around $B^*$ is constructed, as before, to remove the dependency of $\mathcal{C}$ on $\{\tilde{b}_j\}$. However, the cost function still depends on the position of the $\{\hat{b}_k\}$, and $C(B_f)$ is not differentiable at $\{\hat{b}_k^*\} \subset B_f^*$. Let us assume a BP configuration $B_{f,0}$ existed with cheaper cost than $B_f^*$, i.e. $\mathcal{C}(B_f^*) = \mathcal{C}(B_{f,0}) + \Delta$ for some $\Delta > 0$. Lemma C.3 states that we can find an arbitrarily small $\delta$ so that $B_f^* + \delta$ is non-degenerate and the gradient $\nabla_{B_f} \mathcal{C}(B_f^* + \delta)$ is arbitrarily small. Due to the convexity of $\mathcal{C}$, the following inequality holds:

$$\mathcal{C}(B_{f,0}) \geq \mathcal{C}(B_f^* + \delta) + \langle \nabla_{B_f} \mathcal{C}(B_f^* + \delta), B_{f,0} - (B_f^* + \delta) \rangle$$
$$= \mathcal{C}(B_f^*) + \underbrace{O(\|\delta\|_2) + \langle \nabla_{B_f} \mathcal{C}(B_f^* + \delta), B_{f,0} - (B_f^* + \delta) \rangle}_{=: K(\delta)}$$

where we have used that $\mathcal{C}(B_f^* + \delta) = \mathcal{C}(B_f^*) + O(\|\delta\|_2)$, using Big-O notation. All terms which tend to zero as $\delta \to 0$ have been summarized in $K(\delta)$. $K(\delta)$ can be brought arbitrarily close to zero by choosing a sufficiently small $\delta$. In particular, there exists a $\delta > 0$ so that $|K(\delta)| < \Delta/2$. Using the assumption $\mathcal{C}(B_f^*) = \mathcal{C}(B_{f,0}) + \Delta$, this leads to:

$$\mathcal{C}(B_{f,0}) \geq \mathcal{C}(B_f^*) + K(\delta) = \mathcal{C}(B_{f,0}) + \Delta + K(\delta) \geq \mathcal{C}(B_{f,0}) + \frac{\Delta}{2} ,$$

which is a contradiction. Hence $B^*$, is again a local minimum. $\qquad \square$

## D  Properties of the functions $f$ and $h$ describing the optimal branching angles

The optimal branching angles are expressed in terms of the following functions $f(\alpha, k)$ and $h(\alpha, k)$, defined for $\alpha \in [0, 1]$ and $k \in (0, 1)$, cf. Eq. (3):

$$f(\alpha, k) = \arccos\left(\frac{k^{2\alpha} + 1 - (1 - k)^{2\alpha}}{2k^\alpha}\right) ,$$
$$h(\alpha, k) = \arccos\left(\frac{1 - k^{2\alpha} - (1 - k)^{2\alpha}}{2k^\alpha (1 - k)^\alpha}\right) .$$

Figure 13 shows $f$ and $h$ as functions of $k$ for a number of different values of $\alpha$. The two functions are related by $h(\alpha, k) = f(\alpha, k) + f(\alpha, 1 - k)$, so that $h$ is symmetric around $k = 1/2$. Both functions are defined for inputs $\alpha \in [0, 1]$ and $k \in (0, 1)$. For $\alpha = 1$, we have $f(\alpha = 1, k) = 0 = h(\alpha = 1, k)$ for all $k$, reflecting the fact that Y-shaped branchings are never optimal in the optimal transport case. On the other hand, we have $f(\alpha = 0, k) = \pi/3$ and $h(\alpha = 0, k) = 2\pi/3$ for all $k$. This limit corresponds to the Euclidean Steiner tree problem where in an optimal Y-branching all edges meet at $120°$. Moreover, considering the limits of $k \to 0$ and $k \to 1$, one finds that $f(\alpha, k \to 0) \to \pi/2$ and

$f(\alpha, k \to 1) \to 0$. Consequently, $h(\alpha, k) \to \pi/2$ for both $k \to 0$ and for $k \to 1$. In the following three lemmas, we investigate the monotonicities of the two functions.

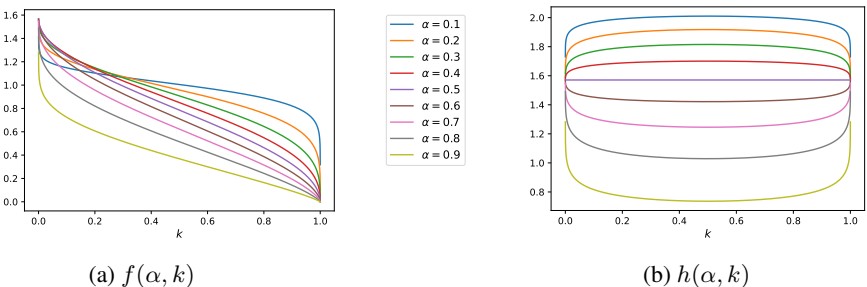

(a) $f(\alpha, k)$  (b) $h(\alpha, k)$

Figure 13: The functions $f$ and $h$ vs. $k$ for different values of $\alpha$.

**Lemma D.1.** $f(\alpha, k)$ *is strictly decreasing in $k$ for all $\alpha \in (0, 1)$ and $f(\alpha, k) < \pi/2$ for all $\alpha \in [0, 1]$ and $k \in (0, 1)$.*

*Proof.* Since the inverse cosine function decreases monotonically, it is sufficient to show that the argument is monotonically increasing. Let us consider its derivative

$$\frac{\partial}{\partial k}\left(\frac{k^{2\alpha} + 1 - (1-k)^{2\alpha}}{2k^\alpha}\right) \sim k^{2\alpha} + (1-k)^{2\alpha} + 2k(1-k)^{2\alpha-1} - 1 =: s(\alpha, k) .$$

The $\sim$ indicates that we have dropped overall factors which were clearly positive, as we are only interested in the sign of the derivative. First, note that $s(\alpha = 0, k) = \frac{1+k}{1-k} > 0$ and $s(\alpha = 1, k) = 0$ for all $k$. Secondly, $s(\alpha, k)$ is strictly decreasing with respect to $\alpha$:

$$\frac{\partial s}{\partial \alpha} = 2k^{2\alpha} \log(k) + 2(1-k)^{2\alpha} \log(1-k) + 4k(1-k)^{2\alpha-1} \log(1-k) < 0 .$$

Each of the above terms are negative due to the fact that $k, (1-k) \in (0, 1)$. Together it follows that $s(\alpha, k)$ is positive for any combination of $\alpha$ and $k$ and the proof of the monotonicity is complete. The fact that $f(\alpha, k \to 0) \to \pi/2$ for all $\alpha$ then immediately implies that $f(\alpha, k) < \pi/2$. □

**Lemma D.2.** *For $\alpha \leq 0.5$, $h(\alpha, k)$ increases monotonically in $k$ for $k \in (0, 1/2]$ and decreases for larger values of $k$. Conversely, for $\alpha \geq 0.5$, $h(\alpha, k)$ increases first until in reaches a maximum at $k = 1/2$ and decreases afterwards.*

*Proof.* The argument here follows the proof of Lemma 12.14 in [2]. Due to the monotonicity of the inverse cosine function, it is again sufficient to investigate the expression in the argument. We rewrite this expression trivially, so that it becomes a function of the fraction $r := \frac{k}{1-k}$ and consider the following derivative:

$$\frac{\partial}{\partial r}\left(\frac{(r+1)^{2\alpha} - r^{2\alpha} - 1}{2r^\alpha}\right) \sim -(r+1)^{2\alpha} - r^{2\alpha} + 1 + 2r(r+1)^{2\alpha-1} =: t(\alpha, r)$$

We first note that $t(\alpha = 0, r) = \frac{r-1}{r+1} \leq 0$ and that $t(\alpha = 0.5, r) = 0$ as well as $t(\alpha = 1, r) = 0$ for all $r$. Next, we show that $t(\alpha, r)$ is a concave function with respect to $r$ for $r \leq 1$, which corresponds to $k \in (0, 1/2]$:

$$\frac{\partial^2 t}{\partial \alpha^2} = 4(r-1)(r+1)^{2\alpha-1} \log^2(r+1) - 4r^{2\alpha} \log^2(r) \leq 0 ,$$

since $r \leq 1$. Taken together, we conclude that $t(\alpha, r) \leq 0$ for all $\alpha \in [0, 0.5]$ and that $t(\alpha, r) \geq 0$ for all $\alpha \in [0.5, 1]$. Moreover, since $r$ increases monotonically w.r.t. $k$, the monotonicity of $h(\alpha, k)$ holds for $k \in (0, 1/2]$. Due to the symmetry of $h$ around $k = 1/2$, the proof is complete. □

**Lemma D.3.** $f(\alpha, k)$ *decreases monotonically in $\alpha$ for $\alpha \in [0.5, 1]$ and $h(\alpha, k)$ decreases monotonically in $\alpha$ for all $\alpha \in [0, 1]$.*

*Proof.* As the inverse cosine function decreases monotonically, it is again sufficient to investigate the derivative of the function in the arccos-argument. For $f(\alpha, k)$, we consider

$$\frac{\partial}{\partial \alpha}\left(\frac{k^{2\alpha} + 1 - (1-k)^{2\alpha}}{2k^a}\right) = \frac{1}{2}k^{-\alpha}\left(\underbrace{\log(k)}_{< 0}[\underbrace{k^{2\alpha} + (1-k)^{2\alpha} - 1}_{\leq 0}] - 2(1-k)^{2\alpha}\underbrace{\log(1-k)}_{< 0}\right) > 0.$$

To see that in fact the expression in the square bracket is smaller or equal to zero, we exploit that for $\alpha \geq 0.5$ the function $k \mapsto k^{2\alpha}$ is superadditive, so that $1 = 1^{2\alpha} = (k+(1-k))^{2\alpha} \geq k^{2\alpha} + (1-k)^{2\alpha}$. For $h(\alpha, k)$, we consider

$$\frac{\partial}{\partial \alpha}\left(\frac{1 - k^{2\alpha} - (1-k)^{2\alpha}}{2k^\alpha(1-k)^\alpha}\right) = -\frac{1}{2}(1-k)^{-\alpha}k^{-\alpha}\left(\underbrace{\log(k)}_{< 0}[k^{2\alpha} + 1 - (1-k)^{2\alpha}]+\right.$$

$$\left.\underbrace{\log(1-k)}_{< 0}[(1-k)^{2\alpha} + 1 - k^{2\alpha}]\right) > 0.$$

Since $k, (1-k) \in (0, 1)$ clearly the expressions in square brackets are positive so that the overall expression is positive too. In the respective regions, $f(\alpha, k)$ and $h(\alpha, k)$ are thus monotonically decreasing in $\alpha$. $\qquad\square$

# E    Non-optimality of higher-degree branchings

This section supplements Sect. 4.1 of the paper. First, we address the third and most involved 4-branching scenario in which a coupled 4-BP connects one source and three sinks (or equivalently 3 sources and 1 sink), see Fig. 14. We derive the inequalities listed in Proposition 4.2 and prove them analytically for a large subset of the parameter space. For the remainder we present a numerical argument (App. E.1.4). Lastly, we show by induction how, given that coupled 4-BPs are never globally optimal, one can further rule out coupled $n$-BPs (with $n$ effective neighbors) for all $n > 4$ (App. E.2).

## E.1    Non-optimality of coupled 4-BPs between one source and three sinks

Let us start by providing the derivation of Proposition 4.2, which we repeat here for completeness:

**Proposition E.1.** *Given a BOT problem with one source and three sinks, with demands $m_1, m_2, m_3$ as in Fig. 14, a coupled 4-BP away from the terminals cannot be globally optimal if at least one of the following inequalities holds true:*

$$\Gamma = h\left(\frac{m_1}{m_1 + m_2}\right) - f(m_1) + h\left(\frac{m_3}{m_3 + m_2}\right) - f(m_3) > 0,$$

$$\Gamma_{1,*} = f(1 - m_*) + f\left(1 - \frac{m_2}{1 - m_*}\right) - f(1 - m_* - m_2) > 0,$$

$$\Gamma_{2,*} = h\left(\frac{m_*}{m_* + m_2}\right) + f\left(\frac{m_2}{1 - m_*}\right) - h(m_*) > 0$$

*where $* = 1, 3$. Note that $\Gamma = \Gamma_{1,1} + \Gamma_{1,3} = \Gamma_{2,1} + \Gamma_{2,3}$.*

Note that it is an important specification that we consider coupled 4-BPs *away from the terminals*. For instance, in the OT case where $\alpha = 1$, all BPs are located at the terminals and coupled BPs with arbitrary number of neighbors may be globally optimal. For all following considerations, we assume that $\alpha \in [0, 1)$.

### E.1.1 Derivation of the $\Gamma$-inequality

WLOG, let us normalize the masses so that $m_1 + m_2 + m_3 = 1$ and determine the necessary conditions under which all V-branchings are optimal:

$$\gamma_1 \geq \pi - f\left(\alpha, 1 - \frac{1-m_1}{1}\right) = \pi - f(\alpha, m_1),$$
$$\gamma_2 \geq \pi - f\left(\alpha, 1 - \frac{1-m_3}{1}\right) = \pi - f(\alpha, m_3),$$
$$\gamma_3 \geq h\left(\alpha, \frac{m_3}{m_3 + m_2}\right),$$
$$\gamma_4 \geq h\left(\alpha, \frac{m_1}{m_1 + m_2}\right). \tag{16}$$

We intend to show that such a 4-BP can never be globally optimal by showing that for any combination of $\alpha$ and the masses $m_i$ the sum of the lower bounds is already larger than $2\pi$. This is equivalent to proving that the following inequality holds true for all parameter combinations:

$$\Gamma := h\left(\alpha, \frac{m_1}{m_1 + m_2}\right) - f(\alpha, m_1) + h\left(\alpha, \frac{m_3}{m_3 + m_2}\right) - f(\alpha, m_3) > 0. \tag{17}$$

The inequality reflects that the problem setup is inherently symmetric under exchange of $m_1$ and $m_3$. WLOG, we assume that $m_1 \leq m_3$.

### E.1.2 Derivation of the $\Gamma_{1,*}$- and $\Gamma_{2,*}$-inequalities

For a BOT solution to be globally optimal it means that it is the cheapest relatively optimal solution of all possible full tree topologies. In our case of four terminals, there are three distinct topologies, see Fig. 15. Let us assume that a globally optimal 4-BP away from the terminals exists and denote the terminal positions by $a_i$. Then, for all three topologies $T_1$, $T_2$ and $T_3$, this branching point configuration is the ROS, since clearly for all $T_i$ a coupled 4-BP configuration can be realized by coupling the two branching points.

Let us investigate graphically under which conditions a coupled 4-BP provides the relatively optimal solution for the different topologies. Figure 16 shows the pivot circle and pivot point construction for topology $T_1$, where $a_0$ has been chosen as root node. Let us refer to the line through $p_2$ and the intersection of the two pivot circles as *transition line*, for the following reason: If $a_0$ was positioned to the left of the transition line, the ROS of $T_1$ would be non-degenerate, as shown in Fig. 4d for instance. For $a_0$ to the right of the transition line, the ROS of $T_1$ is given by a coupled 4-BP. Hence, the transition line marks the transition between a non-generate ROS of $T_1$ and a coupled 4-BP solution. Consequently, $a_0$ must lie to the right of the transition line of $T_1$. But, $a_0$ must simultaneously also lie on the appropriate side of the transition lines of the two other topologies $T_2$ and $T_3$. One can now argue that the root node $a_0$ can be moved along a continuous path onto the transition line of topology $T_1$, without crossing any of the other transition lines. In doing so, the coupled 4-branching stays relatively optimal for all three topologies. Most importantly, it thereby stays globally optimal. During this procedure the terminals $a_1$, $a_2$ and $a_3$ stay fixed so that the pivot points and pivot circles as well as the transition lines stay exactly the same. For topology $T_1$ we have now arrived at a special case of coupled 4-branching, in which the V- and L-branchings are transient (see Def. B.2). Let us refer to such a BP configuration as *transient* 4-branching. As a consequence, the angle $\gamma_4$ may be expressed in terms of the following branching angles (see Fig. 16)

$$\gamma_4 = \beta_1 + \beta_2 - \theta_1 = h(\alpha, m_1) - f\left(\alpha, \frac{m_2}{1 - m_1}\right). \tag{18}$$

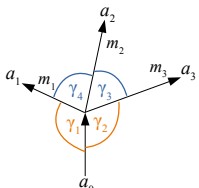

Figure 14: Coupled 4-BP between one source and three sinks.

All in all, the above argument shows that the existence of a globally optimal coupled 4-BP necessarily implies the existence of a globally optimal transient 4-branching, which can be constructed by changing only the coordinates of one of the terminals. Clearly, for the globally optimal and transient 4-BP of topology $T_1$ all four necessary conditions for optimal V-branching must still apply. In order to generally rule out globally optimal coupled 4-BPs, it is therefore sufficient to show that at least one of the following two conditions is fulfilled for all parameter combinations $\alpha$ and $m_i$:

1. the sum of the lower bounds on $\gamma_i$ always exceeds $2\pi$,
2. condition (16) is always incompatible with Eq. (18).

The first condition is equivalent to the following inequality, obtained by substituting the lower bound for $\gamma_4$ in (17) by (18):

$$\Gamma_{1,1}(\alpha, m_1, m_2) := f(\alpha, 1 - m_1) + f\left(\alpha, 1 - \frac{m_2}{1 - m_1}\right) - f(\alpha, 1 - m_1 - m_2) > 0 \,. \quad (19)$$

The second condition can be expressed as inequality simply by combining (16) and (18):

$$\Gamma_{2,1}(\alpha, m_1, m_2) := h\left(\alpha, \frac{m_1}{m_1 + m_2}\right) + f\left(\alpha, \frac{m_2}{1 - m_1}\right) - h(\alpha, m_1) > 0 \,. \quad (20)$$

Proving one of these two inequalities already suffices to rule out globally optimal 4-branching between one source and three sinks. Note that the three inequalities presented so far are not completely independent but are related via $\Gamma = \Gamma_{1,1} + \Gamma_{2,1}$. The above procedure of moving the root node onto the transition line of topology $T_1$ can be repeated exactly analogously for $T_2$. This results in the inequalities $\Gamma_{1,3}$ and $\Gamma_{2,3}$, which are of the exact same form except that $m_3$ appears in all places instead of $m_1$. Note that, in both cases, we have used that $a_0$ can be moved onto the transition lines of $T_1$ and $T_2$ without crossing the transition line of topology $T_3$. A justification for this is given in form of the following lemma.

**Lemma E.2.** *Starting from a globally optimal coupled 4-BP connecting the terminals $\{a_i\}$ and located away from all terminals, one terminal node may be moved along a continuous path onto the transition lines of topology $T_1$ and $T_2$ without crossing the transition line of topology $T_3$ first.*

*Proof.* Let us give a proof by contradiction. Assume that that $a_0$ could actually be moved along a continuous path onto the transition line of topology $T_3$ without touching any of the other two transition lines. Then, one may also move $a_0$ infinitesimally further across the transition line of topology $T_3$, such that the ROS of $T_3$ becomes non-degenerate. At the same time, for topology $T_1$ and $T_2$ the coupled 4-branching configuration stays relatively optimal. Now, since the ROS of $T_3$ deviates from the coupled 4-branching configuration, the 4-branching can no longer be globally optimal. Note that a coupled 4-BP can only be globally optimal if *all three* topologies agree. Consequently, the ROS of $T_3$ must be globally optimal. However, the non-degenerate solution of $T_3$ will necessarily contain a cycle. To see this, let $b_1$ be the branching point connected to $a_0$ and $b_2$ the other branching point in $T_3$. After crossing the transition line of $T_3$, the edge $(b_1, b_2)$ at first has finite but infinitesimal length, so that either one of the edges $(b_2, a_1)$ or $(b_2, a_3)$ must intersect with $(b_1, a_2)$, thereby creating a cycle in the ROS of $T_3$ (cf. Fig. 15c). However, it was proven in [2] that for $\alpha \in [0, 1)$ cyclic BOT solution cannot be globally optimal, so that we have arrived at a contradiction. $\qquad\square$

To summarize, we have shown the following lemma:

**Lemma E.3.** *The existence of any globally optimal 4-BP connecting one source and three sinks and located away from the terminals implies the existence of two globally optimal transient 4-branchings, one for topology $T_1$ and one for $T_2$.*

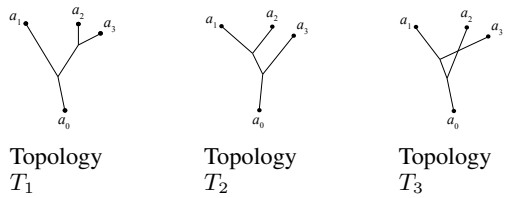

Figure 15: The three distinct full tree topologies connecting four terminals.

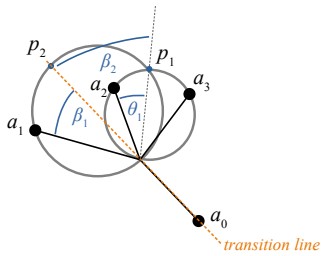

Figure 16: Transient coupled 4-BP with $a_0$ placed on the transition line of topology $T_1$.

### E.1.3 Analytical treatment of the inequalities

**$\Gamma > 0$ for $\alpha \leq 0.5$:**  For $\alpha \leq 0.5$ we have $h \geq \pi/2$ and $f < \pi/2$ for all combinations of $m_i$ (see App. D) and thus clearly $\Gamma > 0$ if $\alpha \leq 0.5$. For all following arguments, we therefore assume that $\alpha > 0.5$. The following analytical arguments all rely on the properties of $h$ and $f$, which are listed and proven in App. D.

**$\Gamma > 0$ for $m_1 \geq 1/4$:**  Using that $f(\alpha, k)$ is monotonically decreasing in $k$, the following loose lower bound suffices to demonstrate that $\Gamma > 0$ if $m_1 \geq 1/4$:

$$\Gamma \geq 2 \min_k h(\alpha, k) - 2f(\alpha, m_1)$$

$$= 2\arccos(2^{2\alpha-1} - 1) - 2\arccos\left(\frac{m_1^{2\alpha} + 1 - (1-m_1)^{2\alpha}}{2m_1^\alpha}\right),$$

where we have used that $f(m_1) \geq f(m_3)$ due to our assumption $m_1 \leq m_3$ and that $h(\alpha, k)$ forms a minimum at $k = 1/2$. For $m_1 = 1/4$, one finds that this expression is truly positive if and only if

$$9^\alpha < 2 \cdot 4^\alpha + 1,$$

which is fulfilled for all $\alpha \in [0, 1)$ due to the subadditivity of the function $m \mapsto m^\alpha$, namely $9^\alpha = (4 + 4 + 1)^\alpha < 4^\alpha + 4^\alpha + 1$. It follows that $\Gamma > 0$ also for all $m_1 > 1/4$ due to the monotonicity of $f$.

**$\Gamma_{2,3} > 0$ for $m_3 \geq 1/2$:**  Inequalities $\Gamma_{2,3}$ is fulfilled if $m_3 \geq 1/2$, since

$$\Gamma_{2,3}(\alpha, m_3, m_2) > h\left(\alpha, \underbrace{\frac{m_3}{m_3 + m_2}}_{> m_3}\right) - h(\alpha, m_3) \geq 0 \text{ for } m_3 \geq 1/2, \tag{21}$$

where we have used that for $\alpha > 0.5$, $h(\alpha, k)$ is monotonically increasing in $k$ for $k \in [0.5, 1)$, as shown in Lem. D.2. The remaining parameter region, for which none of the inequalities have been shown yet, can be characterized by the following conditions:

$$\alpha > 0.5, \quad m_1 < 0.25 \text{ and } m_2 \in [0.5 - m_1, 1 - 2m_1]. \tag{22}$$

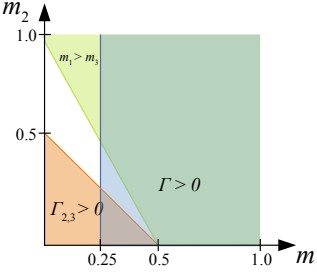

Figure 17: $m_1$-$m_2$ parameter space largely ruled out analytically. In the white region the inequalities are checked numerically.

The constraints for $m_3$ are implicitly represented, using that the normalization was chosen such that $m_1 + m_2 + m_3 = 1$. A visualization of the remaining region can be found in Fig. 17. Finally, we propose a simple and watertight numerical scheme, which can be used to rule out almost the entire remaining volume, already with little numerical effort.

### E.1.4    Numerical treatment of the remaining parameter space

For the remaining parameter region characterized by the conditions in (22), we propose a numerical scheme which checks the inequality $\Gamma_{2,1}$ for all practically relevant parameter combinations of $m_1$, $m_2$ and $\alpha$. For this, we split the remaining volume into cuboids $I = [\check{\alpha}, \hat{\alpha}] \times [\check{m}_1, \hat{m}_1] \times [\check{m}_2, \hat{m}_2]$ which are divided recursively into smaller cuboids based on the following octree scheme. For each cuboid, we determine a lower bound of $\Gamma_{2,1}$. This lower bound becomes tighter the smaller the cuboid $I$. If, for a cuboid, this lower bound is not yet positive, it is divided further into eight new cuboids by splitting each of the three intervals in half. This procedure is iterated until for all cuboids the lower bound is truly positive. For all $m_1, m_2, \alpha$ in a cuboid $I = [\check{\alpha}, \hat{\alpha}] \times [\check{m}_1, \hat{m}_1] \times [\check{m}_2, \hat{m}_2]$, the lower bound of $\Gamma_{2,1}$ is obtained by minimizing each summand individually:

$$\Gamma_{2,1}(\alpha, m_1, m_2) \geq \min_I h\big(\alpha, \frac{m_1}{m_1 + m_2}\big) + \min_I f\big(\alpha, \frac{m_2}{1 - m_1}\big) - \max_I h(\alpha, m_1)$$

$$= h\Big(\hat{\alpha}, \frac{\hat{m}_1}{\hat{m}_1 + \check{m}_2}\Big) + f\Big(\hat{\alpha}, \frac{\hat{m}_2}{1 - \hat{m}_1}\Big) - h(\check{\alpha}, \check{m}_1) \tag{23}$$

where we have used the crucial fact that for the remaining volume, characterized by (22), the functions $h(\alpha, k)$ and $f(\alpha, k)$ are monotonically decreasing in both arguments. The proof of these monotonicities can be found in App. D. Note that the described procedure allows to rigorously confirm the inequality across a continuous region with finitely many evaluations.

For $\alpha \to 1$ or $m_1 \to 0$, the value of $\Gamma_{2,1}$ approaches zero, so that the above scheme cannot be used to proof the inequality for values arbitrarily close to these limits. However, if we restrict us to $m_1 > \delta$ and $\alpha < 1 - \epsilon$ for finite $\epsilon, \delta > 0$, the inequality $\Gamma_{2,1}$ can be shown for practically all parameter combinations with little numerical effort. For $\epsilon = \delta = 10^{-3}$, using the proposed scheme, it was checked in only a few minutes that the lower bound in Eq. (23) is larger than $10^{-4}$ everywhere in the remaining volume. To be numerically on the safe side, we have stopped splitting a cuboid not if the lower bound exceeded zero but set a suitable finite threshold, in this case $10^{-4}$. In other words, we have stopped splitting a cuboid if its respective lower bound was $> 10^{-4}$. The smallest terms which occur during the arithmetic operations inside the functions $h$ and $f$ are of the order $\delta^2$, the largest terms are of order one. It is therefore safe to say that numerical errors at the order of the machine accuracy are negligibly small against the margin of $10^{-4}$ and we may say that all together globally optimal 4-BPs are ruled out, for all practical parameter combinations. In principle, the presented scheme can be used to check the inequality $\Gamma_{2,1}$ up to even smaller $\epsilon$ and $\delta$. The Python code of the numerical scheme is made available at `https://github.com/hci-unihd/BranchedOT`.

### E.2    Non-optimality of five- and higher-degree branchings

In this section, we formally prove that globally optimal coupled $n$-BPs not coincident with a terminal can be ruled out in general, given that 4-BPs are not globally optimal. Lemma 2.1 states that a solution is not globally optimal if any subsolution is not globally optimal. It will therefore suffice to study the coupled BP as an isolated subproblem. Let us start by proving the following corollary about the preservation of relative optimality under edge extensions for transient V- and L-branchings (see Def. B.2):

**Corollary E.4.** *Consider a BOT problem with terminals $A = \{a_i\}$ and let the BP configuration $B = \{b_i\}$ be relatively optimal for a given topology $T$. Let $B$ contain a transient V-branching between two terminals, say $a_0$ and $a_1$ and denote the branching point connected to $a_0$ and $a_1$ by $b_1$. Otherwise, let $B$ not contain any strict L-or V-branchings. Let us further denote the branching point to which $b_1$ is coupled in a transient V-branching by $b_2$, as illustrated on the left side of Fig. 18. Then, there exists a direction in which the zero length edge $(b_1, b_2)$ can be extended to finite length $l > 0$ (cf. right side of Fig. 18) such that the new BOT problem (with shifted terminal positions) is solved relatively optimally by the new BP configuration (with shifted $b_1$).*

*Proof.* By assumption, the BP configuration of interest $B = \{b_i\}$ contains only Y-branchings and transient V- and L-branchings. Then, according to Lemma C.3, there exists a set of arbitrarily small displacements $\delta_i$, one for each branching point $b_i$, so that $B + \delta = \{b_i + \delta_i\}$ is a non-degenerate BP configuration with arbitrarily small gradients $\|\nabla_{b_i} \mathcal{C}(B + \delta, A)\|$. Note that the notation $C(B, A)$ emphasizes that the cost function also depends on the terminal positions. Since the BP configuration $B + \delta$ is non-degenerate, any edge, in our case $(b_1, b_2)$, can be easily expanded in length without changing any of the branching angles, assuming that the extension preserves the direction of the edge and that the BPs and terminals are moved along correspondingly. As the gradient $\nabla_{b_i} \mathcal{C}$ depends only on the directions of the edges meeting at $b_i$ (i.e. the branching angles), the gradient is not changed by this procedure. Let us summarize the shifted BPs by $B_{shift}$ and the shifted terminals by $A_{shift}$. Then, $\|\nabla_{b_i} \mathcal{C}(B + \delta, A)\| = \|\nabla_{b_i} \mathcal{C}(B_{shift} + \delta, A_{shift})\|$ can also be made arbitrarily small as $\delta \to 0$. Let us prove by contradiction that $B_{shift}$ is the ROS of the BOT problem with terminals $A_{shift}$: Assuming that a different BP configuration $B_0 \neq B_{shift}$ was the ROS, there would exist a constant $\Delta > 0$, so that $\mathcal{C}(B_{shift}, A_{shift}) = \mathcal{C}(B_0, A_{shift}) + \Delta$. However, since the cost is a convex function w.r.t. the BPs, we have

$$\mathcal{C}(B_0, A_{shift}) \geq \mathcal{C}(B_{shift} + \delta, A_{shift}) + \langle \nabla_{b_i} \mathcal{C}(B_{shift} + \delta, A_{shift}), B_0 - (B_{shift} + \delta) \rangle$$
$$= \mathcal{C}(B_{shift}, A_{shift}) + \underbrace{O(\delta) + \langle \nabla_{b_i} \mathcal{C}(B_{shift} + \delta, A_{shift}), B_0 - (B_{shift} + \delta) \rangle}_{=:K} .$$

As $\delta \to 0$, the latter two terms summarized by $K$ can clearly be made arbitrarily small. In particular, there exists an $\delta > 0$, such that $|K| < \Delta/2$, which together with $\mathcal{C}(B_{shift}, A_{shift}) = \mathcal{C}(B_0, A_{shift}) + \Delta$ implies that

$$\mathcal{C}(B_0, A_{shift}) \geq \mathcal{C}(B_{shift}, A_{shift}) + K > \mathcal{C}(B_0, A_{shift}) + \Delta/2$$

and we have thereby arrived at a contradiction, similarly to the reasoning in part (c) of the proof in App. C. $\qquad \square$

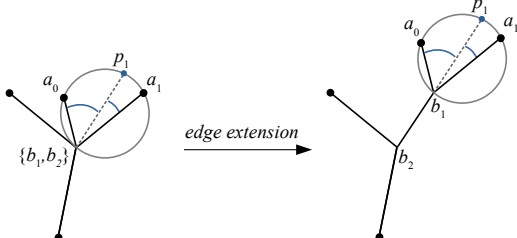

Figure 18: Extension of the edge $(b_1, b_2)$ in the direction of the pivot point $p_1$, preserving relative optimality.

Coming back to the non-optimality of coupled $n$-BPs, let us, for concreteness, consider a globally optimal coupled 5-BP not coincident with a terminal. By induction, repeating the presented argument one can then rule out all globally optimal $n$-BPs. The proof is by contradiction, so let us assume a globally optimal 5-BP existed, between terminals $a_0$, $a_1$, $a_2$, $a_3$ and $a_4$. In terms of enclosed angles $\gamma_i$ (see Fig. 19), it is a necessary condition that all $\gamma_i$ must exceed their respective lower bound, specified by the optimal V-branching conditions in Table 1 (Section 3.2). Again, for a 5-BP configuration to be globally optimal all possible full tree topologies must agree on the 5-BP configuration as their relative optimal solution. Starting from this configuration, one may continuously move one of the terminals, e.g. $a_0$, such that the globally optimal coupled 5-BP starts to *decouple*, meaning that $a_0$ is moved until for (at least) one of the possible topologies, say $\tilde{T}$, the 5-BP configuration no longer provides the ROS. This can always be achieved, for instance by bringing $a_0$ sufficiently close to $a_1$ so that a Y-branching between the two terminals becomes optimal. Similar to the 4-branching case, the ROS of such a topology $\tilde{T}$ in this moment becomes the globally optimal solution, as the other topologies are still in the 5-branching configuration which can only be globally optimal if *all* topologies agree on it. The globally optimal solution of topology $\tilde{T}$ must exhibit one of the following two properties

1. it contains one Y-branching and a coupled 4-BP or

2. it is non-degenerate and contains only Y-branchings.

In the first case, it would mean that a coupled 4-BP exists which is globally optimal as subgraph of a globally optimal solution (using the necessarily optimal substructure of Lem. 2.1). This contradicts our assumption that 4-BPs are not globally optimal.

Regarding the second option, we proceed as follows. Let us move $a_0$ back to the point in which the coupled 5-BP configuration was still globally optimal but the ROS of $\tilde{T}$ is on the verge of decoupling into a non-degenerate branching configuration. Denote this special location of the terminals by $\{a_i^*\}$. In this configuration an infinitesimal movement of $a_0$ away from $a_0^*$ can cause the ROS of $\tilde{T}$ to transition from coupled 5-branching to a non-degenerate BP configuration, very much analogous to the case of the transient 4-branching illustrated in Fig. 16. This means that in this configuration all V- and L-branchings are transient in the ROS of $T$ and WLOG we choose the labeling of the terminals such that the V-branching between $a_0^*$ and $a_1^*$ is transient, cf. Fig. 19. At this point, let us split the set of all possible full tree topologies $\mathcal{T}$ into the following subsets. The subset in which the terminals $a_0$ and $a_1$ are connected to a common branching point, say $b_1$, is denoted by $\mathcal{T}^{(0,1)}$. Let us label the BP to which $b_1$ is connected in these topologies by $b_2$. The ROS of all topologies $T \in \mathcal{T}^{(0,1)}$ for the current BOT problem contains a transient V-branching at branching point $b_1$. Moreover, let us single out a specific subset in $\mathcal{T}^{(0,1)}$, defined by the following condition:

$$\mathcal{T}_{trans}^{(0,1)} = \{T \in \mathcal{T}^{(0,1)} : \text{ROS of } T \text{ is transient if the terminals are located at } a_i^*\} \;.$$

Visibly, the branching at $b_1$ appears as V-branching but it may also be seen as a Y-branching with a zero length stub. Let us now extend this zero length edge between $b_1$ and $b_2$ to finite length $l > 0$ into the direction of the pivot point between $a_0^*$ and $a_1^*$, as explained in the proof of Corollary E.4. The two terminals $a_1$ and $a_2$ are shifted from $a_0^*$ and $a_1^*$ to $a_0(l)$ and $a_1(l)$ and Corollary E.4 guarantees that the resulting BP configuration for all $T \in \mathcal{T}_{trans}^{(0,1)}$ solves the new BOT problem relatively optimally. For an illustration of the edge extension see Fig. 19. Note that for all $T \in \mathcal{T}_{trans}^{(0,1)}$ this ROS is the same. Now split this ROS into two subsolutions as indicted in Fig. 19. This induces two subproblems and subtopologies, as described in Def. C.1. We focus on the upper right subproblem, consisting of four terminals. Note that any topology $T^{(4)}$ on this four terminal subproblem, may be induced as subtopology by a topology $T \in \mathcal{T}^{(0,1)}$. Let us distinguish the following two cases: a) On the four terminal subproblem the topology of the GOS, denoted by $T^{*(4)}$, is induced by a topology $T \in \mathcal{T}_{trans}^{(0,1)}$. Or b) $T^{*(4)}$ is induced by a topology $T \in \mathcal{T}^{(0,1)} \setminus \mathcal{T}_{trans}^{(0,1)}$. In case a) the ROS of $T^{*(4)}$ is given by the right subsolution in Fig. 19, as the edge extension preserved the relative optimality for all $T \in \mathcal{T}_{trans}^{(0,1)}$. Hence, the globally optimal solution on the four terminal subproblem is given by a coupled 4-BP and we have arrived at a contradiction.

Otherwise, in case b), we do not know the ROS of $T^{*(4)}$ a priori, but it cannot be a coupled 4-branching configuration and must hence be non-degenerate. Crucially, it must be non-degenerate for any finite length extension $l > 0$, even if we consider the limit of $l \to 0$. But this means that the ROS of $T^{*(4)}$ already transitions from a coupled 4-BP into a non-degenerate ROS, if the terminals $a_0$ and $a_1$ are perturbed infinitesimally ($l > 0$ but infinitesimal). Consequently, for $l = 0$ the ROS of $T^{*(4)}$ is transient. But this means that $T^{*(4)}$ can be induced as a subtopology of a transient topology

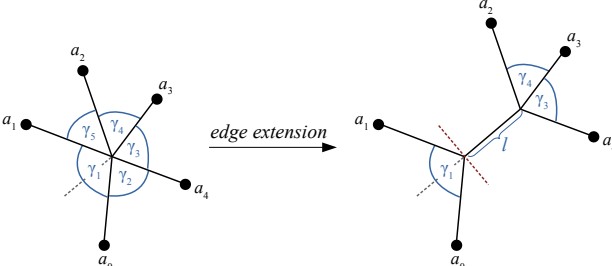

Figure 19: Edge extension to finite length $l > 0$ in a globally optimal and transient 5-branching with transient V-branching between $a_0$ and $a_1$. The resulting ROS is split into two subsolutions, as indicted by the red dashed line.

$T \in \mathcal{T}_{trans}^{(0,1)}$ and we have arrived yet at another contradiction. $\qquad\square$

# F  BOT on two-dimensional Riemannian manifolds

In the following section, we prove that the optimal branching angles for Y-shaped branchings on two-dimensional manifolds are the same as the optimal branching angles derived for BOT in the Euclidean plane. An outline of the proof was given in Section 5.1. The same strategy, presented below for the optimal branching angles, can be used to generalize other necessary conditions for optimal BOT solutions to manifolds, as explained in Sect. F.2.

## F.1  Optimal Y-branching on two-dimensional Riemannian manifolds

For a Y-shaped branching at BP $b$ connecting the terminals $a_0$, $a_1$ and $a_2$ in Euclidean plane the cost function of BOT is given by

$$\mathcal{C}(b) = \sum_i m_i^\alpha \left\| a_i - b \right\|_2 ,$$

where the $m_i$ are the known edge flows. Let us consider a two-dimensional Riemannian manifold $\mathcal{M}$ embedded into $\mathbb{R}^3$, for which the metric is induced by the standard Euclidean inner product in $\mathbb{R}^3$. Let the geodesic distance be denoted by $d(x, y)$. The generalized cost function for 1-to-2 branching then reads

$$\mathcal{C}_M(b) = \sum_i m_i^\alpha \, d(a_i, b) .$$

All points $b$ and $a_i$ now lie on the manifold and are assumed to have differing positions. In a solution which minimizes $\mathcal{C}_M$ the terminals $a_i$ are connected to $b$ via geodesics. For $b$ to be a valid solution to the BOT problem on the manifold, these geodesics must exist. We denote the geodesic which connects $b$ and $a_i$ by $v_i(\lambda) \in \mathcal{M}$, parametrized by the length $\lambda$. The tangent space at $b$ is denoted by $T_b\mathcal{M}$. Furthermore, let $\hat{n}_i$ be the unit tangent vectors of the geodesics $v_i(\lambda)$ at the branching point $b$, i.e. $\hat{n}_i = \partial_\lambda v_i(\lambda)\big|_{\lambda=0}$. WLOG, for all following considerations let us rotate and translate the manifold so that $b \in \mathbb{R}^3$ lies at the origin, i.e. $b = 0$, and that the tangent space $T_b\mathcal{M}$ is equal to the $x_1$-$x_2$-plane of $\mathcal{R}^3$, i.e. $T_b\mathcal{M} = \mathbb{R}^2 \times \{0\}$.

**Restriction to a local subsolution on the manifold.**  Since $\mathcal{M}$ is embedded into $\mathbb{R}^3$, there exists an $r > 0$ and an environment $U(r) \subset \mathbb{R}^3$ around $b = 0$ such that the manifold $\mathcal{M} \cap U(r)$ can be represented as the graph of a function:

$$\mathcal{M} \cap U(r) = \{(x, u(x)) : x \in D(r) \subset \mathbb{R}^2\} ,$$

where $u$ is a smooth, scalar function $u : D(r) \to \mathbb{R}$, defined on the disk of radius $r$, denoted by $D(r) := \{(x_1, x_2) \in \mathbb{R}^2 : \left\|(x_1, x_2)^T\right\|_2 \le r\}$. Note that due to the mentioned rotation and translation of $\mathcal{M}$, we have $u(0) = 0$ and $\nabla u(0) = 0$, where $\nabla u$ denotes the gradient of $u$. The existence of such a function $u$ is guaranteed by the implicit function theorem. A formal proof can be found in John M. Lee's book [18] in Proposition 8.24. Further, let us define the orthogonal projection $\sigma$ from the manifold onto the $x_1$-$x_2$-plane as

$$\sigma : \mathcal{M} \cap U(r) \to D(r) \times \{0\}, \quad \begin{pmatrix} x_1 \\ x_2 \\ u(x_1, x_2) \end{pmatrix} \mapsto \begin{pmatrix} x_1 \\ x_2 \\ 0 \end{pmatrix} . \tag{24}$$

WLOG, $r$ is chosen sufficiently small so that this projection is bijective. Now, Taylor's theorem states that $u$ can be approximated by the following expansion around $x = 0$:

$$u(x) = \underbrace{u(0)}_{= \, 0} + \underbrace{\langle \nabla u(0), x \rangle}_{= \, 0} + O(\|x\|_2^2) \in O(r^2) , \tag{25}$$

where $\langle \cdot, \cdot \rangle$ is the standard Euclidean inner product and we have introduced the Big-O notation. A term is of order $O(r^2)$ if it goes to zero for $r \to 0$ at least as fast as $r^2$, or more formally:

$$p(r) \in O(q(r)) \quad \Leftrightarrow \quad \lim_{r \to 0} \frac{p(r)}{q(r)} = c$$

for some finite constant $c$. Consequently, a point $a = (x, u(x))$ in $\mathcal{M} \cap U(r)$ and its projection onto the plane $\sigma(a)$ agree to first order, i.e.

$$\|\sigma(a) - a\|_2 = u(x) \in O(r^2) \,. \tag{26}$$

One of the key ingredients when transferring BOT problems from two-dimensional surfaces to the tangent plane $T_b \mathcal{M}$ is the following Lemma about the difference between the Euclidean distance and the geodesic distance:

**Lemma F.1** (Relation between geodesic and Euclidean distance). *Let $r$ be a small radius, which characterizes the environment $U(r)$ around the origin $b = 0$ on a two-dimensional Riemannian manifold as described above. Let $a \in \mathcal{M} \cap U$ be a point in this environment, located at $a = (x, u(x)) \in \mathbb{R}^3$ for some $x \in D(r)$ and $u$ as above. Then, the geodesic distance can be expressed through the Euclidean distance as*

$$d(a, b) = \|a - b\|_2 + O(r^3) \,. \tag{27}$$

*Proof.* Let us first note that $d(a, b) \geq \|a - b\|_2$. It is hence sufficient to show that $d(a, b) \leq \|a - b\|_2 + O(r^3)$. Let us consider the following curve $\gamma(t) = (tx, u(tx))$ on the manifold $\mathcal{M}$, which for $t \in [0, 1]$ connects $a = (x, u(x))$ and $b = 0$. In general, $\gamma(t)$ is not a geodesic between $a$ and $b$. Thus, the length of $\gamma(t)$ provides an upper bound to $d(a, b)$. For the calculation of the length we use that $\gamma'(t) = (x, \langle \nabla u(tx), x \rangle)$ and simply integrate $\|\gamma'(t)\|_2$ along the curve in $\mathbb{R}^3$, since the metric of the embedding is induced by the standard Euclidean inner product in $\mathbb{R}^3$:

$$\frac{d(a, b)}{\|a - b\|_2} \leq \frac{1}{\sqrt{\|x\|_2^2 + u^2(x)}} \int_0^1 \|\gamma'(t)\|_2 \, \mathrm{d}t$$

$$\leq \frac{1}{\|x\|_2} \int_0^1 \sqrt{\|x\|_2^2 + |\langle \nabla u(tx), x \rangle|^2} \, \mathrm{d}t$$

$$= \int_0^1 \sqrt{1 + \left| \left\langle \nabla u(tx), \frac{x}{\|x\|_2} \right\rangle \right|^2} \, \mathrm{d}t$$

$$\leq 1 + \frac{1}{2} \int_0^1 \left| \left\langle \nabla u(tx), \frac{x}{\|x\|_2} \right\rangle \right|^2 \, \mathrm{d}t$$

$$\leq 1 + \frac{1}{2} \int_0^1 \|\nabla u(tx)\|_2^2 \, \mathrm{d}t \,.$$

For the last step we have used the Cauchy-Schwarz inequality. Furthermore, we have used that $\sqrt{1 + x^2} \leq 1 + x^2/2$. The only thing left to show is that the remaining integral is $O(r^2)$, as $\|a - b\| = (\|x\|_2^2 + u^2(x))^{1/2} \in O(r)$, since $\|x\|_2 \leq r$ and $u(x) \in O(r^2)$, cf. Eq. (25). For that, let us consider the Taylor expansion of the $i$-th component of $\nabla u(tx)$ around 0. For $i = 1, 2$ one has

$$\partial_i \, u(tx) = \underbrace{\partial_i \, u(0)}_{= \, 0} + \partial_j \partial_i \, u(0) \, tx_j + O(r^2) \,,$$

where the sum in $j$ is implicit and runs over both indices $j = 1, 2$. And thus:

$$\frac{1}{2} \int_0^1 \|\nabla u(tx)\|_2^2 \, \mathrm{d}t = \frac{1}{6} \partial_j \partial_i \, u(0) \, x_j \, \partial_k \partial_i \, u(0) \, x_k + O(r^2) \,,$$

where again the sum over all index pairs is implied. Since for all components $x_i$ we have $x_i \leq r$, this completes the proof. $\qquad \square$

**Projection of a subproblem onto the tangent plane.**   Coming back to the 1-to-2 BOT problem on the manifold, let us project the geodesics $v_i$ onto the $x_1$-$x_2$-plane using the orthogonal projection $\sigma$ from above, in order to transfer the BOT problem from the manifold onto the tangent plane. WLOG, we choose the orientation of the $x_1$ and $x_2$ axis such that $\hat{n}_0$ points along the $x_1$-axis. Then, one may easily check that the implicit function theorem guarantees that for sufficiently small $r$ the projected geodesic $\sigma(v_0)$ can be represented by a graph $(x_1, w(x_1))$ with smooth $w : (-\epsilon, r + \epsilon) \to \mathbb{R}$. Note, that the projected geodesic $\sigma(v_0)$ may be considered on the open interval with $\epsilon > 0$ such that all derivatives with respect to $x_1 \in [0, r]$ are well-defined. Due to our special orientation of the $x_1$-$x_2$-plane, we have $w(0) = 0$ and $w'(0) = 0$. Let us now define the following two points inside the tangent plane:

1. Define $\hat{a}_0$ as the point which lies in the direction of $\hat{n}_0$ at a distance $r$ away from the origin, i.e. $\hat{a}_0 = r \cdot \hat{n}_0 = (r, 0)$. Similarly, define $\hat{a}_i = r \cdot \hat{n}_i$. These will be the three terminals of the BOT subproblem of interest in the Euclidean plane. For an illustration see Fig. 20.

2. Define $a_{\perp,0} = (r, w(r))$ as special point on the projected geodesic $\sigma(v_0)$. From an analogous construction using the projected geodesics $\sigma(v_1)$ and $\sigma(v_2)$, we obtain the $a_{\perp,i}$ also for $i = 1, 2$. No explicit representation of these will be necessary.

What is however important is that the two defined points agree up to linear order, in the sense that $\|\hat{a}_i - a_{\perp,i}\|_2 \in O(r^2)$ for all $i = 0, 1, 2$. This is shown for $i = 0$, by Taylor expansion of $w(x_1)$ around $x_1 = 0$, but holds of course equally true for $i = 1, 2$:

$$\|\hat{a}_0 - a_{\perp,0}\|_2 = w(r) = \underbrace{w(0)}_{= 0} + \underbrace{w'(0)}_{= 0}\, r + O(r^2) \in O(r^2). \tag{28}$$

Note that since the points $a_{\perp,i}$ lie on the projected geodesics, after an inverse projection, the $\sigma^{-1}(a_{\perp,i})$ mark special points on the unprojected geodesics $v_i \in \mathcal{M}$. These will be the terminals of the subproblem of interest on the manifold. Combining Eq. (26) and (28), we can relate $\sigma^{-1}(a_{\perp,i}) \in \mathcal{M}$ to the terminals $\hat{a}_i$ of the BOT problem of interest in the plane by

$$\left\| \sigma^{-1}(a_{\perp,i}) - \hat{a}_i \right\|_2 \in O(r^2) \, , \tag{29}$$

Note that all steps above can be repeated for radii smaller than the chosen $r$. In the following we will decrease the scale $r$ of the two problems and therefore indicate the $r$-dependence in $\sigma^{-1}(a_{\perp,i}(r))$ and $\hat{a}_i(r)$ explicitly.

**Improved subsolutions on the manifold by projection from the tangent space.**   Now, follows the main line of arguments to show that $b$ does not connect $a_i \in \mathcal{M}$ at minimal cost if the geodesic angles in the tangent space $T_b\mathcal{M}$ are different from the optimal angles in the Euclidean plane. We start by considering the subproblem on the manifold with terminals $\sigma^{-1}(a_{\perp,i}) \in \mathcal{M}$. The generalized cost function of this subproblem reads:

$$\mathcal{C}_\mathcal{M}(b) = \sum_i m_i^\alpha\, d(\sigma^{-1}(a_{\perp,i}(r)), b) \, .$$

We can now use Eq. (29) to express the terminals on the manifold through the terminals $\hat{a}_i$ in the tangent plane and express the geodesic distance in terms of the Euclidean distance, based on Lem. F.1.

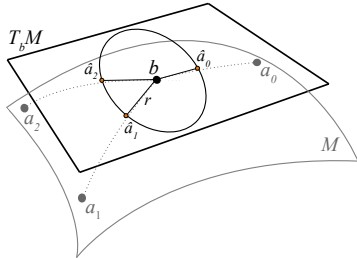

Figure 20: The tangent vectors of the geodesics (dotted) in $b$ define the position of the terminals $\hat{a}_i$ of the subproblem on the flat disk $D(r) \subset T_b\mathcal{M}$.

All additional terms are at least of order $O(r^2)$ and we have

$$\mathcal{C}_M(b) = \underbrace{\sum_i m_i^\alpha \left\| \hat{a}_i(r) - b \right\|_2}_{=: \, \mathcal{C}_b(r)} + O(r^2) \, . \tag{30}$$

Crucially, with $\mathcal{C}_b(r)$ we have arrived at the cost of the BOT problem which purely lives on the tangent space. It is the cost of the solution in which $b$ is connected via straight lines to the terminals $\hat{a}_i(r) = r \cdot \hat{n}_i$ at a distance $r$ away from $b$, as shown in Fig. 20. By assumption, this solution is not relatively optimally since the angles (or equivalently the directions $\hat{n}_i$) are assumed to deviate from the optimal angles in the Euclidean plane. This means that a branching point $b^* \in D(r) \times \{0\}$ exists which provides a better solution with cost denoted by $\mathcal{C}_{b^*}(r) < \mathcal{C}_b(r)$. From the definition of $\hat{a}_i = r \cdot \hat{n}_i$, we see that two BOT problems within disks of different radii $D(r)$ and $D(\nu \cdot r)$ are related simply by rescaling the coordinates. Under rescaling of coordinates the Euclidean distance between points and consequently also the cost functions changes proportionally, i.e. $\mathcal{C} \to \nu \mathcal{C}$. Thus, there exist non-negative constants $M_1$ and $M_2$, so that $\mathcal{C}_b(r) = M_1 \cdot r$ and $\mathcal{C}_{b^*}(r) = M_2 \cdot r$. Let us distinguish the following two cases:

a) $\mathcal{C}_{b^*}(r) = \sum_i m_i^\alpha \left\| \hat{a}_i(r) - b^* \right\|_2 = 0$. This special case looks unusual, but is in principle possible, e.g. if all terminals $a_i$ and $b$ lie on a common geodesic. In the case, where $\mathcal{C}_{b^*}(r) = 0$, we project $b^*$ onto the manifold using $\sigma^{-1}$ and go backwards in the above steps. We find that

$$0 = \sum_i m_i^\alpha \left\| \hat{a}_i(r) - b^* \right\|_2 = \underbrace{\sum_i m_i^\alpha \, d(\sigma^{-1}(a_{\perp,i}(r)), \sigma^{-1}(b^*))}_{= \, \mathcal{C}_M(\sigma^{-1}(b^*))} + O(r^2) \, .$$

Clearly, a sufficiently small $r > 0$ exists so that the terms contained in $O(r^2)$ are much smaller than $M_1 \cdot r$, so that

$$\mathcal{C}_M(\sigma^{-1}(b^*)) = 0 + O(r^2) < M_1 \cdot r + O(r^2) = \mathcal{C}_M(b) \, .$$

This proves that the projection of $b^*$ onto the manifold provides a cheaper cost solution on the manifold than $b$.

b) $\mathcal{C}_{b^*}(r) > 0$, and thus $M_2 > 0$. In this case, let us write the ratio of the two costs as $\mathcal{C}_b(r)/\mathcal{C}_{b^*}(r) = M_1/M_2 =: 1 + \kappa$ for some $\kappa > 0$. Again, $b^*$ and its projection onto the manifold agree to first order, that is $\left\| \sigma^{-1}(b^*) - b^* \right\| \in O(r^2)$. We now show that $\sigma^{-1}(b^*)$ provides a better solution to the BOT problem on the manifold than $b$. We proceed from Eq. (30), using that $\mathcal{C}_{b^*}(r)$ is linear in $r$:

$$\mathcal{C}_M(b) = C_b(r) + O(r^2) = (1 + \kappa) \, \mathcal{C}_{b^*}(r) + O(r^2)$$

$$= \sum_i m_i^\alpha \left\| \hat{a}_i(r) - b^* \right\|_2 + \kappa M_2 \cdot r + O(r^2)$$

$$= \underbrace{\sum_i m_i^\alpha \, d(\sigma^{-1}(a_{\perp,i}(r)), \sigma^{-1}(b^*))}_{= \, \mathcal{C}_M(\sigma^{-1}(b^*))} + \kappa M_2 \cdot r + O(r^2) \, .$$

In the last step, we have projected inversely onto the manifold and replaced the Euclidean distance by the geodesic distance (all with differences at least of $O(r^2)$). We have therefore arrived at the cost of the new solution on the manifold. Since the term $\kappa M_2 \cdot r$ is positive, we conclude that $r$ may always be chosen sufficiently small, so that the linear term dominates over higher-order terms and we have:

$$\mathcal{C}_M(\sigma^{-1}(b^*)) < \mathcal{C}_M(b) \, .$$

Together with Lem. 2.1 on the necessarily optimal substructure of all subsolutions, this concludes the proof that any optimally placed branching point on a two-dimensional embedded Riemannian manifold must exhibit the same optimal branching angles as in the Euclidean plane. $\qquad\square$

### F.2 Other local properties of optimal BOT solutions on manifolds

The logic of the proof outlined above can be easily extended to the conditions for optimal V- and L-branching. For BOT problems in the Euclidean plane, Table 1 lists these conditions under which the

optimal BP position in a 1-to-2 branching coincides with one of the terminals. Let us now consider a 1-to-2 BOT solution on a two-dimensional manifold in which $b$ is located at the terminal $a_0$ and is connected to the terminals $a_1$ and $a_2$ via the geodesics $v_1$ and $v_2$ respectively. Imagine that in the tangent plane $T_b\mathcal{M}$ the angle enclosed by these geodesics does not fulfill the optimal V-branching criterion. In this case, one may again consider a sufficiently small disk of radius $r$ around $b = a_0$ in the tangent space $T_b\mathcal{M}$ and project the corresponding subproblem from the manifold onto this disk. In the Euclidean plane, an improved BP location $b^*$ must exists, since $b = a_0$ is the optimal solution if and only if the V-branching condition is fulfilled. All arguments about the scaling of the cost improvement apply as described above and the projection $b^*$ back to the manifold will provide an improved solution to the subproblem on the manifold if $r$ is chosen sufficiently small.

Even more so, the same reasoning can also be applied to generalize our results regarding the non-optimality of higher-degree branchings. Given a coupled $n$-BP at position $b$ not coincident with a terminal, one may again consider a sufficiently small region around $b$ on the manifold and project the corresponding subsolution onto $T_b\mathcal{M}$. Exactly analogous to the previous arguments, improving the topology locally in the plane and projecting back to the manifold eventually results in an improved solution on the manifold.

### F.3   The practical side of BOT on manifolds

Although much of the theory generalizes nicely to Riemannian manifolds, the generalization of the practical algorithms is highly non-trivial. Unlike in Euclidean space, realizing the optimal branching angles is a necessary but no longer sufficient condition for relatively optimal solutions. For instance, on the sphere the meridians of three terminals located in the southern hemisphere at longitudes $0°$, $120°$, $240°$ will intersect at both poles at angles of $120°$, which is the optimal angle for $\alpha = 0$. Nonetheless, only the south pole is the optimal branching point. In essence, the geometry optimization aims to assign simultaneously to each branching point the coordinates of the weighted geometric median of its neighbors, a problem that is considered in [8]. The topology optimization presented in Sect. 6.2 could be easily generalized to manifolds if the geodesic distance can be computed. Due to these obstacles, previous works of the Steiner Tree problem ($\alpha = 1$) have focused mostly on the sphere as important special case [7].

## G   Algorithms

**Hardware and code availability.**   Python code for all experiments can be found at `https://github.com/hci-unihd/BranchedOT`. For the different experiments, a single machine with 56 CPUs (Intel(R) Xeon(R) CPU E5-2660 v4 @ 2.00GHz) and 256GB RAM was used. Execution times for all described experiments lie (at most) in the order of hours. A more detailed estimate can be obtained from the performance statistics reported in Fig. 23 and Fig. 26.

### G.1   Geometric construction of relatively optimal solutions for BOT with multiple sources

In Sect. 3.2 we have presented the the exact geometric construction of relatively optimal solutions based on [2, 9] and its generalization to the case of BOT with multiple sources. We have thereby solved the open problem 15.11 in [2], for which an example is illustrated in Fig. 21.

Figure 21 shows a simple BOT problem with a chosen topology for which no root node can be chosen such that all branchings are symmetric (see Sect. 3.2). Choosing for instance the left source as a root node, the branching at $b_1$ is symmetric whereas the one at $b_2$ is asymmetric. However, after having solved the asymmetric branching case analogously to the symmetric one, the relatively optimal solution can now be constructed geometrically as shown in Fig. 22d. More examples of the geometric construction for a given topology applied BOT problems with multiple sources are shown in Fig. 22.

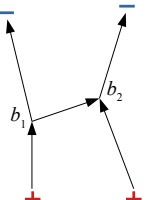

Figure 21: Simple setup with symmetric branching and asymmetric branching, see Sect. 3.2.

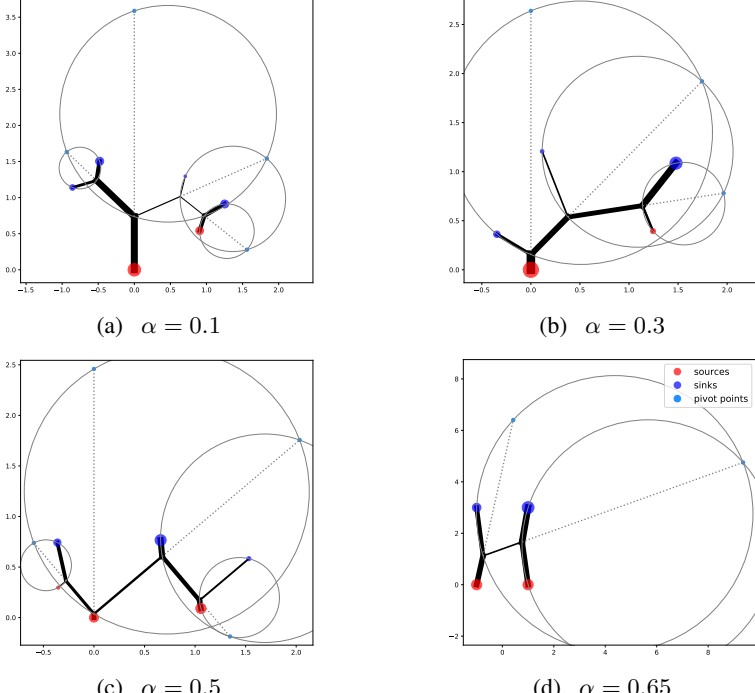

(a)  $\alpha = 0.1$

(b)  $\alpha = 0.3$

(c)  $\alpha = 0.5$

(d)  $\alpha = 0.65$

Figure 22: Examples for geometric construction of relatively optimal solutions with fixed topology for BOT problems with multiple sources. The construction is based on the optimal branching angles and uses one pivot circle and pivot point (light blue) per branching point. Sources are shown in red, sinks in blue.

**Continuity of the optimal BP configuration.**   The limitations of the geometric construction above are discussed in Sect. 3.2. Still, it forms the basis for our theoretical arguments in the paper. Moreover, from the construction of a single branching point based on the branching angles (see Fig. 3b), it can be seen that its optimal position changes continuously w.r.t. the neighbor positions, the edge flows or $\alpha$. Namely, the Y-, V- and L- branching change continuously as the source $a_0$ is moved around. In addition, the optimal branching angles are given by continuous functions of $k$ and $\alpha$ (cf. Eq. (3)), making the construction of the pivot point and pivot circle continuous. By transitivity and based on the optimal substructure in Lem. 2.1, the continuity generalizes also to the construction of larger ROS for a given topology.

## G.2   Numerical algorithm for geometry optimization

In this section, we provide theoretical details and practical experiments for the numerical geometry optimization presented in Sect. 6.1. We have generalized this approached from the context of the ESTP [26] to BOT.

For a given tree topology $T$, the BP configuration is optimized by minimizing the following cost function:

$$\mathcal{C}(X) = \sum_{(i,j)\,\in\,T} m_{ij}^{\alpha} \left\| x_i - x_j \right\|_2 , \tag{31}$$

where the $x_i$ for $1 \leq i \leq n$ are fixed terminals and the $x_i$ for $n+1 \leq i \leq n+m$ denote the variable branching point positions. All coordinates are collectively summarized by $X$.

Starting from a non-optimal, non-degenerate BP configuration denoted $X^{(0)}$, e.g. from a random initialization of the branching points, the algorithm iteratively solves the following *linear* system of equations

$$x_i^{(k+1)} = \sum_{j\,:\,(i,j)\in T} m_{ij}^{\alpha} \frac{x_j^{(k+1)}}{|x_i^{(k)} - x_j^{(k)}|} \Bigg/ \sum_{j\,:\,(i,j)\in T} \frac{m_{ij}^{\alpha}}{|x_i^{(k)} - x_j^{(k)}|}, \quad \text{for } n+1 \leq i \leq n+m. \tag{32}$$

In essence, this is an iteratively reweighted least squares (IRLS) approach [4]. To see this, let us rewrite the cost function in Eq. (31) into pseudo-quadratic from:

$$\mathcal{C}(X) = \sum_{(i,j)\,\in\,T} m_{ij}^{\alpha} |x_i - x_j|^{(\beta-2)+2} = \sum_{(i,j)\,\in\,T} \underbrace{m_{ij}^{\alpha} |x_i - x_j|^{\beta-2}}_{=:w_{ij}(X)} |x_i - x_j|^2 , \tag{33}$$

with $\beta = 1$ for BOT. Clearly, the weights $w_{ij}$ depend on the branching point positions. However, the idea of IRLS is to insert the coordinates $X^{(k)} = \{x_i^{(k)}\}$ at $k$-th iteration into $w_{ij}$ to obtain a truly quadratic form:

$$Q^{(k)}(X) = w_{ij}(X^{(k)})|x_i - x_j|^2 = \sum_{(i,j)\,\in\,T} m_{ij}^{\alpha} \frac{|x_i - x_j|^2}{|x_i^{(k)} - x_j^{(k)}|} ,$$

where we have plugged in $w_{ij}(X^{(k)})$ and $\beta = 1$. Indeed, minimizing this quadratic form yields $X^{(k+1)}$ as in Eq. (32). The updated coordinates $X^{(k+1)}$ are then plugged into $w_{ij}(X)$ during the next iteration. This iterative updating of the weights $w_{ij}$ gives IRLS its name.

Smith [26] proved in detail that this iterative solver converges to the minimum cost BP configuration. The reasoning in [26] is based on the following key argument: As $X^{(k+1)}$ is the minimizer of $Q^{(k)}(X)$, surely $Q^{(k)}(X^{(k)}) \geq Q^{(k)}(X^{(k+1)})$. Together with the fact that $\mathcal{C}(X^{(k)}) = Q^{(k)}(X^{(k)})$, we have

$$\mathcal{C}(X^{(k)}) = Q^{(k)}(X^{(k)}) \geq Q^{(k)}(X^{(k+1)})$$

$$= \sum_{(i,j)\,\in\,T:} m_{ij}^{\alpha} \frac{\left(|x_i^{(k)} - x_j^{(k)}| + |x_i^{(k+1)} - x_j^{(k+1)}| - |x_i^{(k)} - x_j^{(k)}|\right)^2}{|x_i^{(k)} - x_j^{(k)}|}$$

$$= \mathcal{C}(X^{(k)}) + 2[\mathcal{C}(X^{(k+1)}) - \mathcal{C}(X^{(k)})]$$

$$+ \sum_{(i,j)\,\in\,T} m_{ij}^{\alpha} \frac{\left(|x_i^{(k+1)} - x_j^{(k+1)}| - |x_i^{(k)} - x_j^{(k)}|\right)^2}{|x_i^{(k)} - x_j^{(k)}|} .$$

Since the sum in the last expression is clearly non-negative, the inequality above implies that $\mathcal{C}(X^{(k)}) \geq \mathcal{C}(X^{(k+1)})$. This means that the cost of the BP configuration decreases with each iteration which, as shown in [26], implies that the iterations defined by Eq. (7) converge to an absolute minimum of $\mathcal{C}(X)$.

As can be seen from the above derivation, the edge flows $m_{ij}^{\alpha}$ are constant coefficients which do not complicate the considerations in [26]. Consequently, all arguments presented there can be directly transferred from ESTP to BOT. For a detailed discussion on the complexity of the algorithm and suitable convergence criteria, we refer the reader to the work of Smith and only briefly state the results here. Based on an analytically tractable example, Smith claims that the algorithm requires at most $O(N/\epsilon)$ iterations to converge to a solution of the ESTP whose cost is within $\epsilon$ of the optimal cost. As a convergence criterion, Smith suggests to stop the iteration when the angles at all branching points are sufficiently close to the optimal angle conditions. In order to be able to apply the algorithm also

to trees with higher-degree branching points where the optimal angle conditions are a necessary but not sufficient condition for the cost minimum, the experiments in this paper use a different criterion. The algorithm is considered to have converged if from one iteration to the next the relative cost improvement $\left(\mathcal{C}(X^{(k)}) - \mathcal{C}(X^{(k+1)})\right)/\mathcal{C}(X^{(k+1)})$ has dropped below a certain threshold. The BP optimization routine for a given tree topology is summarized in Alg. 1. Note that, in practice, the denominators $|x_i^{(k)} - x_j^{(k)}|$ in Eq. (32) are clipped to $10^{-7}$ to avoid numerical instabilities.

---

**Algorithm 1** BP optimization routine

---

**Input:** threshold $\eta$, tree topology $T$, BOT problem (terminal positions $x_{1:n}$, supplies/demands $\mu$, $\alpha$)
**Output:** numerical minimal cost BP configuration

1: $F \leftarrow get\_all\_edge\_flows(T, \mu)$       ▷ uniquely determined from flow constraints
2: $x_{n+1:n+m} \leftarrow$ randomly initialized.
3: $C_{old} \leftarrow \infty$
4: $C \leftarrow BOT\_cost(T, x_{1:n+m}, F, \alpha)$
5: **while** $\frac{C_{old} - C}{C} > \eta$ **do**
6:    $C_{old} \leftarrow C$
7:    $x_{1:n+m} \leftarrow update\_BPs(x_{1:n+m}, F, \alpha)$     ▷ solves linear system in Eq. (32)
8:    $C \leftarrow BOT\_cost(T, x_{1:n+m}, F, \alpha)$
9: **end while**
10: **return** $x_{1:n+m}$

---

**BOT with costs scaling non-linearly with the edge length.** Let us briefly consider BOT with a modified cost function of

$$\mathcal{C}(X) = \sum_{(i,j) \in T} m_{ij}^\alpha \, \|x_i - x_j\|_2^\beta \tag{34}$$

including one additional parameter $\beta$, regulating the scaling of the cost function w.r.t. edge lengths. For $\beta \geq 1$ the cost function as a function of the branching points is still convex, as $x \mapsto x^\beta$ is convex and increasing and the Euclidean norm is convex. Thus, it has a unique minimum. As can be seen from Eq. (33), the IRLS scheme completely absorbs the $\beta$ into the weights $w_{ij}$. Consequently, the geometry optimization as presented here is readily applicable also to the modified cost in Eq. (34). However, for $\beta > 1$ we are not aware of any theoretical convergence guarantee to the minimum, therefore, further investigation is required. Though, in the special case of $\beta = 2$, it is clear that the geometry optimization can be solved to global optimality in just a single iteration.

**Performance test of numerical BP optimization.** In order to evaluate how long the BP optimization for a given topology takes, we have randomly generated 1000 BOT problems for different number of terminals using Alg. 2. For each of these problems, we have applied the BP optimization routine, given a uniformly sampled full tree topology. Regarding the generation of problem setups, note that BOT solutions and problems are invariant under rescaling of the total demand and supply as well as under global rescaling of the coordinates, see Eq. (1). WLOG, these scales are chosen to be 1. Figure 23 shows the average runtime in seconds plotted against the number of terminals. For BOT problems with 10 to 1000 terminals our efficient C++ implementation of the geometry optimization takes just a fraction of a second. The relatively large error bars, indicating the standard deviation, are not due to an insufficient sample size but due to the natural run time variability for different problems that exists independent of the $n$-dependence. For instance, plotting the average runtime of all problems with $\alpha \leq 0.5$ and $\alpha > 0.5$ separately reveals that the optimization on average requires more time for problems with higher $\alpha$. The reason for this was not investigated further, but we suspect that for larger $\alpha$ on average more V- and L-branchings occur which may require more iterations than Y-branchings for convergence. For this and all following experiments we chose the convergence threshold $\eta$ to be $10^{-6}$.

**Scaling of the geometry optimization** A single iteration of the geometry optimization, i.e. solving the linear system defined by Eq. (7) once, takes $O(nd)$ operations for a problem with $n$ terminals in $d$ spatial dimensions. The elimination scheme used in our efficient C++ implementation is based on the "elimination on leaves of a tree" found in [26]. From Eq. (7), one can easily see that the geometry

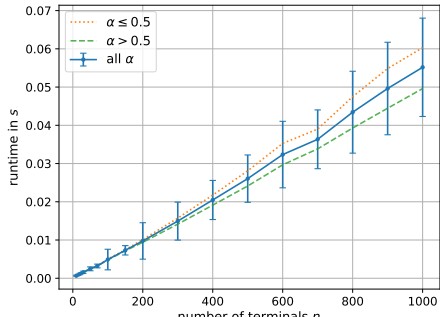

Figure 23: Average runtime of BP optimization routine applied to each 1000 random BOT problems of different size. Average of all problems with $\alpha \leq 0.5$ in orange (dotted) and $\alpha > 0.5$ in green (dashed).

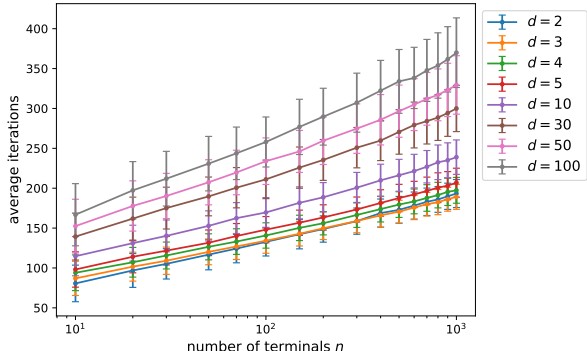

Figure 24: Average number of iterations required by the BP optimization routine until convergence for each 1000 random BOT problems of different sizes $n$ and different dimensionality $d$. The x-axis is in log-scale. The plot suggests that, for the investigated regime of $n$ and $d$, the number of iterations on average scales like $\log(n)$, though one cannot generalize this statement.

optimization parallelizes over the spatial dimensions, as the linear system of the form $Ax = b$ shares the same matrix $A$ across the different dimensions and only the $b$ is different. Although a single iteration is of order $O(nd)$, it is a priori not clear how many iterations are required until convergence is reached. Paralleling the setup of Fig. 23, we have conducted an experiment where we report the number of iterations, for a batch of BOT problems. We performed the same experiment in Euclidean space of different dimensions ($d \in \{3, 4, 5, 10, 30, 100\}$). Figure 24 shows the number of iterations until convergence plotted against the number of terminals. The plot suggests that, for the investigated regime of $n$ and $d$, the number of iterations on average scales like $\log(n)$. However, this is merely an empirical observation and theoretical investigations are an interesting subject for future research.

---

**Algorithm 2** Random BOT problem generation

---

**Input:** number of terminals $n$
**Output:** BOT problem with $n$ terminals located in $[0,1] \times [0,1]$, total supply and demand equal to 1

1: $\alpha \sim \text{Uniform}([0,1])$        $\triangleright$ sample uniformly from $[0,1]$
2: $n^+ \sim \text{Uniform}(\{1, 2, ..., n-1\})$       $\triangleright$ number of sources
3: $n^- \leftarrow n - n^+$
4: $\mu^+_{1:n^+} \sim \text{Uniform}([0,1], \text{size}=n^+)$      $\triangleright$ array of supplies
5: $\mu^-_{1:n^-} \sim \text{Uniform}([0,1], \text{size}=n^-)$      $\triangleright$ array of demands
6: $a_{1:n} \sim \text{Uniform}([0,1], \text{size}=(n,2))$     $\triangleright$ terminal coordinates
7: **return** $a_{1:n}$, $\mu^+/\text{sum}(\mu^+)$, $\mu^-/\text{sum}(\mu^-)$, $\alpha$

---

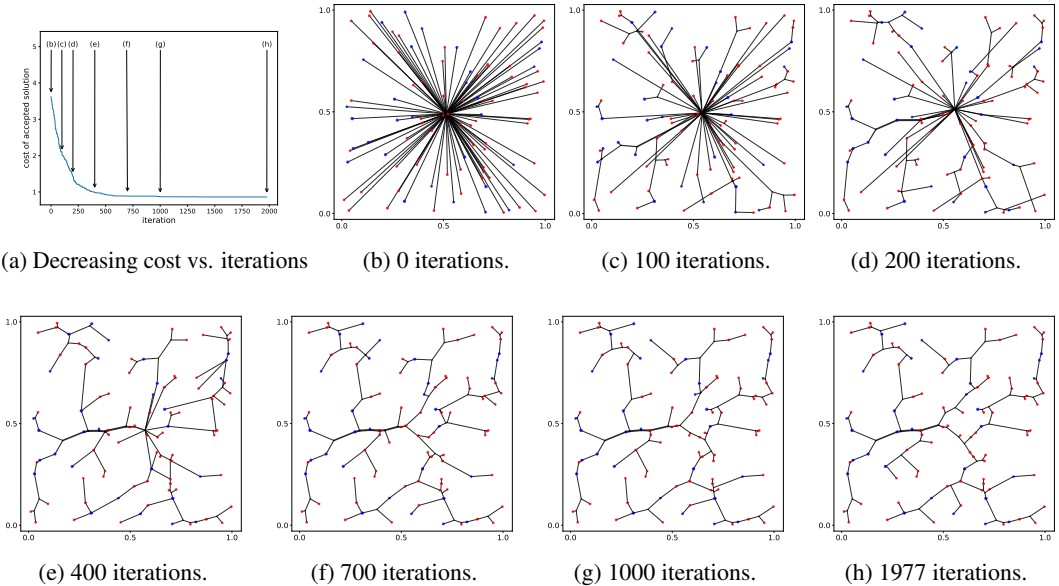

(a) Decreasing cost vs. iterations    (b) 0 iterations.    (c) 100 iterations.    (d) 200 iterations.

(e) 400 iterations.    (f) 700 iterations.    (g) 1000 iterations.    (h) 1977 iterations.

Figure 25: Our greedy BOT solver applied to a problem with 100 terminals (sources in red and sinks in blue), $\alpha = 0.58$.

### G.3   Greedy randomized heuristic for topology optimization

In this section, we give the details of the greedy randomized heuristic presented in Section 6.2 and show the results of some additional experiments conducted with it.

Starting from a tree topology $T$, a uniformly sampled edge $\hat{e}$ is removed. The node in the smaller connected component $\ell$ is connected via a new branching point to an edge in the other component. This edge is sampled according to $p(e) \propto \exp(-d(e,\ell)^2/d_{min}^2)$, where $d(e,\ell)$ is the distance between an edge $e = (i,j)$ and node $\ell$, defined by

$$d(e,\ell) = \min_{\lambda \in [0,1]} \|[x_i + \lambda(x_j - x_i)] - \ell\|_2 \tag{35}$$

and $d_{min}$ is the distance to the closest considered edge. The resulting new tree topology $T_{new}$ is accepted if it decreases the cost compared to the previous topology $T$ and the above procedure is iterated. This greedy optimization strategy is summarized in Alg. 3 below.

Figure 25 shows an example BOT problem with 100 terminals to which our greedy heuristic has been applied. As starting point, the topology with the least possible structure was used, a star-like tree centred around a single BP of degree 100, cf. Fig. 25b. It can be seen how the topology evolves over less than 2000 iterations to the final solution. Figure 25a shows the decreasing transportation cost $\mathcal{C}$ plotted against the number of iterations.

**Performance test of heuristic topology optimization.**   Finally, in a large experiment, we have investigated how many iterations the greedy heuristic requires on average to converge. For that, the greedy heuristic was applied to a number of BOT problems with $n$ terminals and the number of iterations until convergence were counted. The mean and standard deviation of the required iterations are plotted in Fig. 26. The reason for the relatively large standard deviations are due to intrinsic variation of the sampled problems. To illustrate, for instance, the influence of the $\alpha$-value, the average number of iterations until convergence were plotted separately for all BOT problems with $\alpha \leq 0.5$ and $\alpha > 0.5$. We find that the greedy heuristic systematically needs more iterations to converge for $\alpha > 0.5$.

**Influence of the edge sampling kernel.**   Our heuristic topology optimization involves a number of design choices which may affect its performance. A systematic investigation is beyond the scope of this work and left for future research. However, one hyperparameter of particular interest is the kernel (chosen to be Gaussian) and its width (chosen to be $d_{min}$), which together define the replacement

**Algorithm 3** Greedy randomized heuristic (zero temperature limit)

**Input:** BOT problem $P$, tree $T$ interconnecting terminals $a_{1:n}$ (with help of BPs)
**Output:** heuristic BOT solution

1: $B \leftarrow optimize\_BP\_configuration(T, P)$ ▷ returns coordinates of BPs and terminals, see Alg. 1
2: $\mathcal{C} \leftarrow BOT\_cost(T, B, P)$
3: $E \leftarrow \text{list(T.edges())}$
4: **while** $E$ not empty **do**
5:     $S \leftarrow$ Store current state $(T, B)$
6:     $\mathcal{C}_{old} \leftarrow \mathcal{C}$
7:     $\hat{e} \sim \text{Uniform}(E)$                                            ▷ sample uniformly from $E$
8:     $E.\text{remove}(e)$
9:     $T.\text{remove\_edge}(\hat{e})$
10:    $C_b \leftarrow$ subgraph component of $T$ with more nodes
11:    $E_b \leftarrow \text{list}(C_b.\text{edges})$
12:    $b \leftarrow$ node of $\hat{e}$ in $C_b$
13:    $\ell \leftarrow$ node of $\hat{e}$ not in $C_b$
14:    **if** degree$(b)$ == 2 **then**
15:        $n_1, n_2 \leftarrow neighbors(T, b)$
16:        $T.\text{remove\_node}(b)$                      ▷ remove unnecessary BPs with degree 2
17:        $T.\text{add\_edge}(n_1, n_2)$
18:        $E_b.\text{remove}((n_1, b), (n_2, b))$
19:    **end if**
20:    $d \leftarrow$ Initialize array of size $length(E_b)$
21:    **for** edge $e \in E_b$ **do**
22:        $d[e] \leftarrow get\_distance(e, \ell)$            ▷ calculate distance defined by Eq. (35)
23:    **end for**
24:    $d_{min} \leftarrow \min(d)$
25:    $e_c \sim \exp(-d^2/d_{min}^2)$                 ▷ sample edge according to distance kernel
26:    $T.\text{add\_node}(b_{new})$                              ▷ Initialize a new BP
27:    $T.\text{add\_edges}((\ell, b_{new}), (b_{new}, e_c[0]), (b_{new}, e_c[1]))$
28:    $B \leftarrow optimize\_BP\_configuration(T, P)$
29:    $\mathcal{C} \leftarrow BOT\_cost(T, X, P)$
30:    **if** $\mathcal{C} < \mathcal{C}_{old}$ **then**
31:        $E \leftarrow \text{list(T.edges())}$                         ▷ accept new state
32:    **else**
33:        $T, B \leftarrow S$                            ▷ Restore old state $(T, B)$
34:        $\mathcal{C} \leftarrow \mathcal{C}_{old}$
35:    **end if**
36: **end while**
37: **return** $T, B$

probability of the edges, see l. 25 in Alg. 3. To study its influence on the performance we varied the width of the Gaussian kernel $\exp(-d^2/(\omega d_{min})^2)$ by tuning the parameter $\omega$. Based on 150 random problems of various sizes $n$, we calculated the average cost ratio of the heuristic with different $\omega$ to the default of $\omega = 1$ (cf. Fig. 27a). Indeed, Fig. 27a shows that the default choice of $\omega = 1$ is quite strong and relatively robust given that $\omega = 0.5$ or $\omega = 0.1$ work similarly well. Clearly, wider kernels lead to larger (i.e. less local) changes of the topology. At later stages of the algorithm most of these topology changes will be unfavorable, explaining why the algorithm for wider kernels terminates with comparatively less optimal solutions. Furthermore, we have investigated the influence of the kernel width on the number of iterations required (cf. Fig. 27b). Qualitatively, Fig. 27b confirms that for wider kernels, which encourage exploration, more iterations are required. Fitting a power law of the form $x \mapsto ax^b$ to the curves in Fig. 27b, one finds that, depending on the kernel, the greedy heuristic on average scales between $O(n^{1.5})$ and $O(n^{1.7})$. Again, this is a purely empirical statement. A careful theoretical analysis to obtain guarantees, also for problem sizes not covered in our experiments, is beyond the scope of this work.

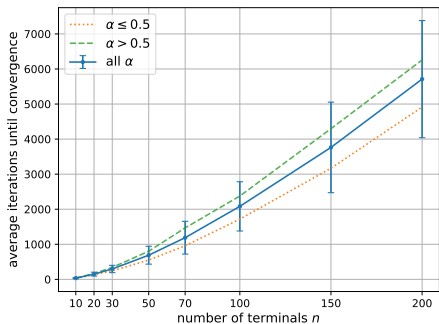

Figure 26: The greedy topology optimization applied to each 150 problems with $n$ terminals: Mean and standard deviation of required iterations plotted vs. $n$ in blue. Average for problems with $\alpha \leq 0.5$ in green (dashed) and for problems with $\alpha > 0.5$ in orange (dotted).

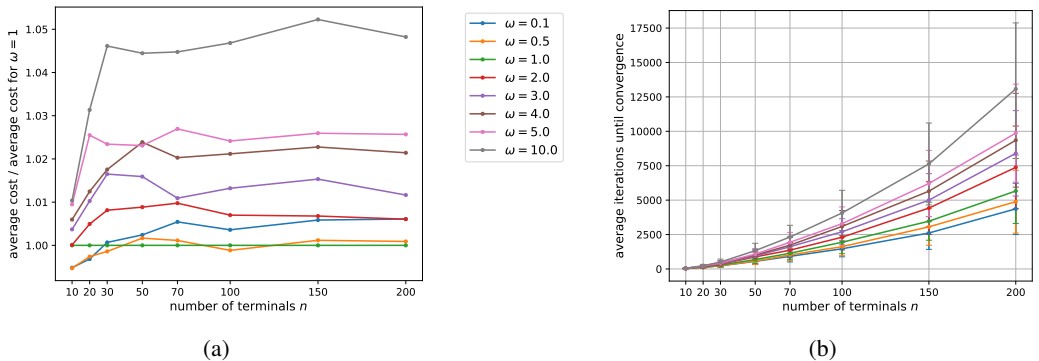

(a)                                                   (b)

Figure 27: Influence of kernel width in topology optimization based on 150 random problems of various sizes $n$: **(a)** Ratio of the average cost of the heuristic solution with kernel width factor $\omega$ to the default of $\omega = 1$. **(b)** Average number of iterations until convergence for different kernel width factors $\omega$. Depending on the kernel width the number of iterations on average scales between $O(n^{1.5})$ and $O(n^{1.7})$.

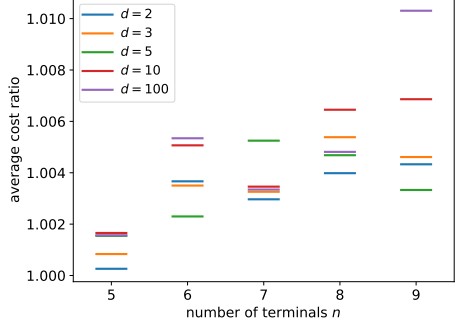

Figure 28: Average cost ratios of our greedy heuristic and brute-force solutions (the closer to 1 the better) for different number of terminals $n$ and dimensions $d$. For each $n$, we uniformly sampled 100 different BOT problems. Though the average cost ratio increases slightly with $n$, our approximate BOT solver compares very well against the ground truth solutions, independently of the dimensionality $d$.

**Greedy topology optimization vs. brute-force search for higher-dimensional BOT.** Both the numerical geometry optimization and the greedy algorithm for the topology optimization presented in Sect. 6 are readily applicable to BOT problems in $\mathbb{R}^d$. Paralleling the experiment presented in

Fig. 8, we have compared our heuristic topology optimization against brute-forced solutions for each 100 problems of size $n = 5$ to $n = 9$ and spatial dimension $d$. Although Fig. 28 suggests that the average cost ratio increases slightly with $n$, our approximate BOT solver again compares very well against the ground truth solutions, independently of the dimensionality $d$.