# OpenReview forum: "Theory and Approximate Solvers for Branched Optimal Transport with Multiple Sources"
_NeurIPS.cc/2022/Conference — NeurIPS 2022 Accept_

### Official Review · Reviewer_mpG7 · 2022-07-07

**Rating:** 6
**Confidence:** 4
**Soundness:** 4 excellent
**Presentation:** 3 good
**Contribution:** 3 good

**Summary:**

The paper under review studied the problem of branched optimal transport with multiple sources.
As a generalization of OT, BOT optimizes the transportation by allowing extra branching points.
A hyperparameter $\alpha$ is used to control the efficiency of transporting mass together.
The transportation flow is captured as a graph, whose vertex set contains the sources, sinks and BP.
In order to obtain an optimal solution,
one needs to find both how vertices are connected (topology) and where the BPs are located (geometry).

Under the above framework, the authors proved that for a graph to be optimal,
the degree of a branch point has to be 3.
Moreover, the authors provided an analytic approach and a numerical algorithm to find the optimal locations of BP.
Furthermore, the authors presented an algorithm to approximate the optimal topology of BOT.
In addition, the authors considered generalization to 2-D Riemannian manifolds.


**Questions:**

Q1: Regarding the topology of $T$: it is unclear what information are enclosed in the topology of $T$.
From the description in the paper, it seems only the graph structure is consider as the topology of a tree.
However from the set up (eq 1), number of branch points ($|B|$), weights on edges ($m_{ij}$) are all variables.
How does one optimize these. In particular, for instance if two edges going out of a source, how does one
determine how much flow each edge shall carry?



Q2: (continuation of Q1) In line 106, the authors stated "first, we determine all edge flows (see Sect. 2)".
How are the edge flows are chosen? Could not find in section 2. Are the edge flows randomly initialized,
then optimized somehow?


Q3: Line 289: "Then, one calculates the distance $d(e, c)$
between $c$ and every edge $e = (i, j)$ in the larger component". How does the distance $d(e, c)$ is defined here?

Q4: Some interpretation on "no degree greater than 3 branch points
are possible for optimal solution" would be nice.
If it is just graph, one can easily switch between
a degree 4 branch point and two degree 3 branch points.
It is not feasible when cost are related,why?

Q5: Is the optimal solution for BOT unique?



Minor points:
- change of disk sizes in Fig 1 is not that visible, could scale a bit more.

- it is probably better to move algorithms to the main text if space allows.

- line 293, "node c" --> "component c".


**Limitations:**

The authors mentioned that it is difficulty
to verify the efficiency of the proposed algorithm for large BOT problem.
My guess is that the running time will be long for the proposed BOT problem comparing to OT.
Considering BOT is a NP-problem, it is expected. It would be nice if the authors could say a bit
more along this line.

**Strengths And Weaknesses:**

Strengths:
The paper is well-written and easy to follow.
The idea of allowing multiple sources for BOT is natural,
the interpolation between OT and euclidean Steiner tree problem is interesting.
The results are novel and sound.


Weaknesses:

A little concern regarding the motivation and practicality.

Motivation: it would be nice if the authors could provide some further explanation why would BOT be useful.
For example, describe a practical example where combined transportation is feasible,
then in that example, how do one pick $\alpha$ to model the case (illustrate that OT is not optimal).
And further how to use algorithms (or theoretical results) developed in this paper to obtain an optimal plan.

Practicality: the running time is not shown in the paper, how fast is the algorithm comparing to Sinkhorn for OT,
i.e. for a given problem, pick $\alpha = 1$, $\alpha = 0.5$, run Sinkhorn for $\alpha = 1$,
run Alg 1 and Alg3 (from the supp) for $\alpha = 0.5$, comparing the running time.

---

> ### Author Response · Authors · 2022-08-02
> **Response to reviewer mpG7**
>
> Thank you very much for your positive review and the detailed feedback! We will address the comments not covered in the overall response here.
>
> Regarding answers to the following topics, please have a look at the corresponding part of our overall response:
> + Runtime of practical algorithms
> + Comparing our solver against existing approaches
>
> **1 A real-world example for BOT**
>
> Many practically relevant real-world problems can be formulated as instance of BOT, such as water supply, mail delivery and other problems related to resource allocation. Let us also give a concrete example to illustrate how one may choose the $\alpha$-parameter. For instance, when designing a water distribution network, the construction cost of pipes may be proportional to the amount of material used, i.e. proportional to their length and diameter. Since the flow through a pipe is proportional to the diameter squared, conversely, the cost is proportional to the square root of the flow, i.e. $\alpha = 1/2$. The choice of $\alpha$ in general relies on a careful inspection of the problem at hand.
>
> **2 Relation between topology and edge flows**
>
> As explained in Sect. 2, WLOG, the topology is chosen to be a full tree topology that has $n-2$ branching points (BPs), each of degree 3. Each of the $n$ terminals must then have degree 1. This does not limit the generality of our explored solutions as higher-degree BPs can form effectively if two or more BPs are located at the same position (as illustrated in Fig. 2).
> Furthermore, for _any_ tree topology the edge flows are already uniquely determined by the flow constraints in Eq.(1) (i.e. supply, demand and flow conservation). The constraints form a linear system of equations which has a unique solution. In Sect. 2 (around l. 67), we state that this system can be solved in linear time, by an approach called ``elimination on leaves of a tree'' in [23].
> Let us for a full tree topology explain very briefly how this elimination works: Given a full tree topology, choose any branching point that is connected to two terminals. The flows through the two terminal edges are known from their respective demand or supply. The flow through the third edge joint at this branching point is thus immediately obtained from flow conservation. On the tree defined by the remaining edges, this procedure is repeated until all flows are determined.
>
> **3 Clarification on coupled branching points**
>
> As in the previous answer, WLOG, the topology of a BOT solution can be represented as full tree topology, in which every branching point (BP) has degree 3. Indeed, a degree-4-branching may be realized by two branching points coupling in the same location (see Def. of coupled BPs in Sect. 2). Regarding a final BOT solution, a BP that truly has degree 4 (on the graph level) and a coupled BP with effective degree 4 are fully equivalent. Therefore, by showing that coupled BPs located away from terminals cannot be globally optimal, we show the same for solutions which include degree-4-BPs on the graph level. Regarding the interpretation of our result, we do not have a satisfactory answer why diversifying a flow in 3 directions at the same time is never optimal. One can only conjecture that the subadditivity of BOT forces flow to accumulate, which penalizes branching and that therefore bifurcations are delayed as far as possible.
>
> **4 Distance between edge and "connecting" node**
>
> In the greedy heuristic, after an edge has been cut, one of the nodes incident to the cut edge is connected to a new edge. This new edge needs to be chosen from the opposite connected component, such that afterwards the topology is connected again. We should clarify that the node which forms this new connection to an existing edge is denoted by $c$ (for "connecting node"). $c$ does here not refer to the whole component. The distance from the node $c$ to an edge $e$ is simply given by the distance between a point and a line segment (see Eq. (33) in appendix G.3). This will be clarified in the final version of our manuscript.
>
> **5 Are optimal BOT solutions unique?**
>
> In general, the globally optimal solution to a BOT problem is not unique. This is most easily illustrated with a highly symmetric counterexample. Picture a BOT problem with two sinks of equal demand and two sources of equal supply positioned on the corners of a perfect square. The sources should be on opposite corners (similar to the setup in Fig. 6(a)). In this case two parallel horizontal paths from sources to sinks are an equally good solution as two vertical ones. In problems with little symmetry, it is however expected to be rather rare that two different transport plans have exactly the same cost.

---

> > ### Comment · Reviewer_mpG7 · 2022-08-07
> > **Thank you for the response.**
> >
> > Thank you for addressing my concern on motivation and practicality, and clarified the work flow of the topology selection.

---

### Official Review · Reviewer_C23R · 2022-07-09

**Rating:** 4
**Confidence:** 3
**Soundness:** 3 good
**Presentation:** 3 good
**Contribution:** 3 good

**Summary:**

The paper studies Branched Optimal Transport (BOT). The problem consists of finding the optimal topology, and achieving the optimal geometry of the BOT network of multiple sources and sinks, given the optimal topology. While it is known that the optimal topologies for such embeddings are trees, the authors make a further step by show that, in the optimal topology, every branching point has at most 3 neighbors. Given the optimal topology, this work also proposes a new geometric optimization strategy for the case of multiple sources, which is generalized from an approach in the literature.  The results also generalize to BOT on Riemannian manifolds. Finally, based on the developed results, they propose heuristic algorithms to solve the BOT problem.


**Questions:**

N.A.

**Limitations:**

As above.

**Strengths And Weaknesses:**

My main concern is that the paper's scope might not be much relevant to the ML community, while certain technical contribution can be interesting. The theoretical results can fit the mathematics community more for its combinatorial and geometrical nature, or fit the OR community for its relevance to routing and/or transportation systems. Yet, I cannot see how this paper can much benefit the ML community. Note that though OT is a simpler formulation than that considered in this work, it has applications in GAN vastly, domain adaptation, and color transfer, etc. so research aiming to solve OT competitively or study OT has been helpful.
The authors may consider well motivating the paper by some concrete applications in ML, whereby BOT is part of the problem formulation or at least conveys some idea/intuition.

---

> ### Author Response · Authors · 2022-08-02
> **Response to reviewer C23R**
>
> Thank you very much for your review! In our overall response, we present a number of arguments why we consider BOT a topic of interest for the ML community. We identify different concrete examples for which current approaches and challenges in ML can benefit from our work and hope to thereby address your concerns regarding the scope of our work adequately.

---

### Official Review · Reviewer_ooE5 · 2022-07-11

**Rating:** 5
**Confidence:** 3
**Soundness:** 3 good
**Presentation:** 2 fair
**Contribution:** 2 fair

**Summary:**

The authors perform a theoretical study of the branched optimal transport (BOT) in the plane, and also suggest a heuristic optimization strategy for the problem. On the theoretical side, they give a geometric construction algorithm for plane BOT solutions, argue for degree 3 branching of the optimal networks via a generalized argument of Bernot et al., and generalize this statement to 2D Riemannian manifolds embedded in $R^3$.

On the numerics and optimization side, their algorithm has two subalgorithms: one aimed at optimization of branching points given a tree topology, and another aimed at optimizing the tree topology. The first is inspired and drawn from a method of Smith for finding Steiner trees, while the second follows an informed heuristic based on node-edge distance to delete random edges and replace them with improved topologies. The method is tested on small numbers of terminals and is shown to achieve optimality quickly in many instances compared to the brute-force approach.

**Questions:**

1. Do you have any specific machine learning applications in mind for this problem? Many examples are given or suggested for OT with convex costs, but not for OT with concave costs. In particular, I imagine the plane restriction might be rather limiting, since most data in ML is high-dimensional.
2. Do you have more detailed information on runtime that could be placed in the main text and/or summarized there? I think many researchers looking to apply your method would be interested to have them there.
3. Do you have any insights on whether the techniques or heuristics would be valid for the problem in higher dimensions?
4. Why didn't you compare to the algorithm of Oudet and Santambrogio (A Modica-Mortola approximation for
branched transport and applications)?

**Limitations:**

The authors have acknowledged the limitations of their method, which mainly center around limited scope, but have not motivated why this limited scope is still interesting.

**Strengths And Weaknesses:**

Strengths:

- The theoretical arguments are rather extensive and seem valid, as far as I have checked them.
- The problem tackled is an interesting one, from a theoretical perspective, and does not seem to be extensively studied.
- The result on limited degree of branching points is a nice, easy to understand, and impactful result.

Weaknesses:
- The paper does not cover or recommend any concrete ML applications of the plane BOT problem.
- Nearly all of the theoretical arguments and constructions, as well as the optimization procedure for fixed topology are generalizations of ideas presented in other approaches.
- The result on generalization to 2D embedded surfaces seems relatively straightforward and not all that significant perhaps. Especially without algorithms adapted to that scenario, or example uses.
- The exposition is challenging to follow, as it generally provides brief glimpses into various theoretical constructions, and then directs the reader to extensive appendices.
- The numerical section is rather brief, given the applied nature of NeurIPS. Only one simple experiment is referenced (though more are in the appendix.

This paper leaves me wandering if it would be better split into a theoretical and numerical investigation. The theoretical portion could be submitted to a different venue, while the numerical portion could be more extensively explored and detailed. As it stands, the paper is sort of an odd omnibus.

---

> ### Author Response · Authors · 2022-08-02
> **Response to reviewer ooE5**
>
> Thank you very much for your detailed review and helpful remarks! Below, we provide responses to comments not addressed in our overall response.
>
> Regarding answers to the following topics, please have a look at the corresponding part of our overall response:
> + Relevance of BOT to ML community and concrete applications
> + Runtime of practical algorithms
> + Structuring our theoretical and practical contributions
> + Comparing our solver against existing approaches
>
> **1 Generalization of techniques and algorithms to higher-dimensional BOT**
>
> **Theoretical results for higher-dimensional BOT.**
> Much of the theory developed in our paper generalizes straightforwardly to BOT problems defined in higher-dimensional Euclidean space. Firstly, optimal BOT solutions are acyclic also in higher dimensions. Furthermore, in Euclidean space, the geometry optimization is convex irrespective of the dimensionality of the problem. Thus, the optimal substructure of BOT solutions covered in Lemma 2.1, which is pivotal in our theoretical reasoning, applies to BOT in $\mathcal R^d$. Additionally, the optimal branching angles, defining the optimal position for branching points of degree 3 (see Eq. (3)), generalize, simply because three points always lie on plane also in higher dimensions.
> However, the recursive construction for BOT solutions (presented in Sect. 3.2) does not generalize, as the pivot point degeneracy mentioned in the end of Sect. 3.2 gets substantially worse for BOT problems in higher dimensions. Admittedly, also the results on the degree limitation do not generalize, as the arguments rely on the fact that the angles between edges meeting at a coupled branching point sum up to $2 \pi$ (cf. Sect. 4.1).
>
> **Practical algorithms for higher-dimensional BOT.**
> The numerical geometry optimization as well as the greedy algorithm for the topology optimization presented in Sect. 6 are readily applicable to BOT problems in $\mathcal R^d$. Moreover, in the course of the rebuttal, we have conducted extra experiments that seem to indicate that the runtime (see overall answer 2 on runtimes) as well as the quality of the solution achieved by our algorithm does not depend strongly on the dimensionality of the problem. Regarding, the quality of the obtained solutions, we repeated the experiment reported in Fig. 8 (comparison against brute force solutions), for higher dimensions (3,5,10,100). Independent of the dimensionality, our approximate solver again compared very well against the ground truth solutions. These experiments will be included in the final version of our manuscript.
>
> We strongly agree with the reviewer that the generalization to higher dimensions is of great interest for various applications of BOT. We will make sure to put more emphasis on the mentioned aspects in the revised version of our paper.
>
>
> **2 Most arguments and constructions are generalizations**
>
> Science is incremental and therefore naturally builds on existing, approved methods. Our work on the generalization of many existing approaches is an important contribution to unify them in a larger framework, which we see as a strength of our work, not as a weakness.
>
>
> **3 Is our theoretically oriented work a good fit for NeurIPS?**
> We agree with the reviewer that the emphasis of our work lies on the theory. However, amongst the most influential NeurIPS papers of the past years [I], we find several works which are largely theoretical in thrust and were appreciated by the community.
>
> [I] https://www.paperdigest.org/2021/02/most-influential-nips-papers/

---

> > ### Comment · Reviewer_ooE5 · 2022-08-08
> > **Thank you for detailed commentary**
> >
> > Thank you very much for the extended responses. They've certainly been very informative, and I've also appreciated the improvements made in the updates. It was especially heartening to understand the authors' efforts in comparison attempts.
> >
> > I would like to keep my score as is however, as I still feel that the paper would be best split into a theoretical portion and a numerical/applied portion. The detailed arguments of the theoretical portion would be of little interest, I imagine, to potential users with the suggested applications in mind. Likewise, those interested in proving structural results about BOT may not care to know about numerical details or algorithms, or potential applications. A separate theoretical paper could be written more cohesively, without the need to push arguments to the appendix and to squeeze things into 9 pages. A numerical/applied paper could expand upon potential applications and provide more extensive experimentation and results within one or multiple use cases.
> >
> > Ultimately, I do want to note that my estimation of the work has increased, and I would not be unhappy if it is accepted as is.

---

### Official Review · Reviewer_CZsf · 2022-07-20

**Rating:** 7
**Confidence:** 3
**Soundness:** 4 excellent
**Presentation:** 3 good
**Contribution:** 4 excellent

**Summary:**

The authors consider the branched optimal transport (BOT) problem with multiple sources. There is a finite number of sources with supplies and sinks (terminals) with demands located at fixed positions. We optimize the transportation between sinks and demands with respect to a directed edge-weighted graph and positions of nodes. The edges interconnect the terminals with the help of additional nodes (branching points). The edge direction indicates the direction of mass flow. The edge weight specifies the absolute flows.

First, the authors investigate the topology and geometry of BOT solutions. They prove sufficient and necessary conditions for a BOT solution to be optimal for a chosen topology (lemma 2.1). Then the authors proposed a geometric approach to optimize BOT solutions, a generalization of the previously proposed approach, developed for BOT problems with a single source.

Second, the authors consider conditions under which coupled branching points (BPs) are not optimal, which can be used to improve the transportation cost of a BOT solution. In particular, they proved that topology could be improved if its relatively optimal solution (ROS) contains coupled BPs.

Third, they consider the case of two-dimensional Riemannian manifolds embedded into 3-dimensional Euclidean space. The authors proved a necessary condition for a ROS.

Finally, they propose heuristics and numerical optimization, which contains an effective algorithm for geometry optimization, followed by a heuristic for topology optimization. The authors claim that the paper is the first to propose heuristics for multiple sources, which do not require user supervision.

**Questions:**

- The paper (if we also consider the results in the Appendix) is very long. Still, I think it could be interesting to see some corner cases:
1) The authors mentioned methods from [26, 28] for the single source case. What is about comparing their performance to the proposed approach? Also, a comparison with the method from [21] can be made.
2) For alpha = 1, we get an OT problem. What is about comparing the proposed approach to some standard OT solvers?
3) Performance of the simulated annealing procedure significantly depends on how the probability of sampling a particular edge is defined and how the normalizing factor d_min is defined. Any empirical results on how this influences the performance?

- Description of Algorithm 2 in Appendix: step 7 is missing?

- On the one hand, the theoretical part of the paper is well written; on the other hand, I think some additional structuring of the text is needed. The paper proposes a novel algorithm. Different parts of the algorithm are scattered across the main text, while their detailed description is in the Appendix. I would propose to include some description of the main steps of the algorithm and references to the corresponding sections of the main text and the Appendix where the detailed description is provided.

- Figure 3: it seems that in Figure 3(c) a_0 is missing

- Section E.1.4 about the “numerical” proof for the remaining parameter space. Any comments on specific technical difficulties why it is impossible to provide analytical proof?

- Any fundamental qualitative explanation of the magic scaling n^{1.4} in line 1089 in the Appendix? What is the reason for such scaling?

- Any experiments for the Riemannian case?

**Limitations:**

The paper does not have any negative societal impact. It can have even positive societal impact since some resource allocation problems can be solved using the proposed approach, which is essential from ESG perspective.

I think that the limitations of the proposed approach are also adequately addressed.

**Strengths And Weaknesses:**

Strengths
- The authors consider a problem statement of a branched optimal transport (BOT) problem with multiple sources. Although the problem statement is purely theoretical, many real-world problems can be formulated in this way. So I consider the topic of the paper important.
- The results seem novel: the paper contains many new results about the properties of ROS for the BOT problem and new algorithms.
- The text of the paper is clearly written.

Weaknesses
- The paper is a bit theoretical. In particular, the authors did not consider solutions to any applied problems based on the proposed methods.
- I would propose to add some additional ablation studies; see comments below.

---

> ### Author Response · Authors · 2022-08-02
> **Response to reviewer CZsf**
>
> Thank you very much for your positive review and your helpful feedback! In the following, we provide responses to your comments.
>
> Regarding answers to the following topics, please have a look at the corresponding part of our overall response:
> + Structuring our theoretical and practical contributions
> + Comparing our solver against existing approaches
> + The practical side of BOT on manifolds
>
> **1 Details on the greedy topology optimization**
>
> In our greedy topology optimization (see Sect. 6.2), at each iteration the current best tree topology is modified by deleting an edge and replacing it with a new one. There are a number of design choices involved in this process. We obtained excellent results with choices that we considered intuitive. But surely this field is wide open for additional experimentation, which we consider beyond the scope of this already substantial work. Still, we report a few additional experiments here:
>
> **Influence of the edge sampling kernel.**
> Indeed, one hyperparameter of particular interest is the kernel (chosen to be Gaussian) and its width (chosen to be $d\_{min}$), which together define the replacement probability of the edges. To study its influence on the performance we varied the width of the Gaussian kernel $\exp(-d^2/(\omega \, d\_{min})^2)$ by tuning the parameter $\omega$. Based on 150 random problems of various sizes ($n \in \\{10, 20, 30, 50, 70, 100, 150, 200 \\}$), we calculated the average cost ratio of the heuristic with given $\omega$ to the default of $\omega=1$:
>
> |  |  |  |  |  |  |  |  |  |
> |--------------|-----------|------------|--|--|--|--|--|--|
> |**kernel width $\omega$**| 0.1     | 0.5        | 1 | 2 | 3 | 4 | 5 | 10 |
> |**average cost ratio**     | 1.002 | 0.999     | 1.0 | 1.006 | 1.012 | 1.018 | 1.023 | 1.041 |
>
> In fact, the default choice of $\omega = 1$ is quite strong and relatively robust given that $\omega = 0.5$ or $\omega = 0.1$ work similarly well. Clearly, wider kernels lead to larger (i.e. less local) changes of the topology. At later stages of the algorithm most of these topology changes will be unfavorable, explaining why the algorithm terminates with comparatively less optimal solutions.
>
> **Scaling of the greedy algorithm.**
> In Section 6.2 we report that the average number of iterations until convergence roughly scales as $O(n^{1.4})$, where $n$ is the number of terminals in the BOT problem. This is merely an empirical observation but no theoretical guarantee. The number of iterations depends on the stopping criterion, the initial topology guess, the kernel defining the sampling of replacement edges and other design choices. Within the experiment described in the above paragraph, we could confirm that for wider kernels which encourage exploration more iterations were required. Consequently, for wider kernels the number of iterations scales roughly as $O(n^{1.7})$ rather than $O(n^{1.4})$.
>
>
> **2 The difficulty of the analytical proof**
>
> We cannot rule out the existence of a fully analytical proof to the inequalities listed in Prop. 4.2. However, the main obstacle in the analytical treatment arises from the nature of the $\arccos$-function (see Eq. (3)) and its derivatives, which are hard to treat analytically. Fortunately, the analytical results obtained by our careful reasoning were just enough to formulate a suitably tight lower bound (Eq. (23)) for $\Gamma_{2,1}$ that enables the use of a feasible numerical scheme.
>
> **3 Clarification on Figure 3(c)**
>
> Figure 3(c) shows a stand-alone illustration of the central angle property used in the recursive construction (cf. Sect. 3.2). The central angle theorem requires only two points. The two nodes where denoted $a\_1$ and $a\_2$ to emphasize the usage of the central angle theorem in the construction.
>
> **4 Small remark on algorithms**
>
> All algorithms in the appendix have an empty line prior to the return statement. We thank the reviewer for the keen eye and will remove these empty lines in the final version.

---

> > ### Comment · Reviewer_CZsf · 2022-08-09
> > **comment to response**
> >
> > I am satisfied with the response to my review. So I can keep the score as it is.

---

### Author Response · Authors · 2022-08-02
**Overarching comments (1 of 3)**

We thank all reviewers cordially for the time invested in reviewing our paper and appreciate their helpful comments. We are very happy to read that the reviewers find that the topic of our paper is important (CZsf, ooE5), that our contributions are novel, sound and impactful (ooE5, mpG7, C23R) and that our paper is clearly written (CZsf, mpG7). We will address questions raised by multiple reviewers here and answer to individual concerns in separate replies. Numbered citations refer to the references of the paper, while alphabetical citations refer to new references.


**1 Relevance of BOT to ML community and concrete applications (ooE5, C23R)**

We all agree that "standard" optimal transport (OT) is by now an important tool in ML [1, 4, 20]. At the same time, routing problems have become a popular problem to challenge machine learning and amortized optimization algorithms with difficult optimization problems [3, 15]. In our eyes, branched optimal transport (BOT) provides the ML community with an interesting model, a very difficult challenge and an important additional tool:

- **Optimizing BOT engenders non-trivial structure**: Many machine learning problems such as tracking of divisible targets (computer vision), skeletonization (image analysis), trajectory inference (bioinformatics) come with input that is essentially continuous (images, distributions) and require structured output that is discrete (lineage trees, cyclic and acyclic graphs). To our mind, the very transition from continuous to discrete is one of the most interesting aspects (and an unsolved problem) in current machine learning research. It is also a problem that cannot be solved by a mere upscaling of standard deep learning architectures. Now, BOT offers a mathematical formalism that is deceivingly simple (a one-liner objective function, plus constraints on mass conservation, see Eq. 1) and yet engenders non-trivial structure as soon as the exponent $\alpha$ becomes smaller than one. We believe that BOT can be a highly instructive toy problem for machine learning, and maybe more: a future tool in the community's box.

Below we speculate on future use-cases, the first being one that we are actively looking into:

- **ML for Science: single cell transcriptomics**:
A challenging task in single-cell transcriptomics is to study the temporal evolution of cells developing from stem cells into highly specialized cells. Already popular approaches exist based on standard OT, such as [A] which states that cells behave like "trains moving along branching railroad tracks", but which yields only diffuse assignment fields. Biologists are interested in sparser abstractions of the raw data, which we believe BOT might provide.

- **ML for Science: evolutionary optimization**:
Unsurprisingly, a great variety of biological systems practically solve BOT in ambient space such as slime mold (Physarum polycephalum) [B], neurons, vasculature or ant colonies. It is interesting to see i) how well BOT approximates these systems, ii) how well these very different systems solve BOT, and iii) to estimate the systems' respective $\alpha$-exponent which might afford new insights into the cost tradeoffs these systems have to implement.

- **ML for Science: robotics**:
For path planning in adverse conditions, swarms of robots or drones may choose to travel together. This line of work has nontrivial ethical concerns, which is why we abstain from it.

- **BOT for hierarchical assignment between two spatially ordered sets**: One of the most popular applications of standard OT is that of measuring the dissimilarity between two sets of points/distributions. While OT only studies the individual assignment of each element, BOT agglomerates the trajectories of different elements. Thus, BOT, in addition to the transportation cost, yields natural clusters on both sets (at no extra cost), revealing a more structured relation between them. The parameter $\alpha$ regulates the level of coarseness of the correspondence (see Fig. 1).

On the question of NeurIPS scope: Optimization (convex, non-convex) is one of NeurIPS areas, according to the call for papers. We introduce an interesting problem that requires both, convex and non-convex optimization, along with its analysis and an optimization heuristic. More importantly, we expect BOT to be of potential interest not only in operations research but also in other fields including biology, environmental science and social sciences. We hope that our illustrations tellingly convey that BOT is of interest to the ML community. In the final version of our manuscript we will emphasize further the relation of BOT with ML by stressing the aforementioned points.

_References_
[A] Schiebinger, Geoffrey, et al. Optimal-transport analysis of single-cell gene expression identifies developmental trajectories in reprogramming. (2019)
[B] Kramar, M., Alim, K. Encoding memory in tube diameter hierarchy of living flow network. (2021)

---

> ### Author Response · Authors · 2022-08-02
> **Overarching comments (3 of 3)**
>
> **4 BOT on Riemannian manifolds (CZsf, ooE5)**
>
> We find that the generalization of BOT to curved surfaces is not extensively studied yet, despite being interesting both from a theoretical and practical perspective. Our results go beyond existing work in the field of the Steiner tree problem [E] and our approach can be used to further generalize optimality criteria from Euclidean space to manifolds (see appendix F.1). Regarding example use-cases of BOT on manifolds, the most important example is surely BOT on the sphere to model transportation on planet earth. Furthermore, for potential applications of BOT in data science, it is a very desirable property to be able to restrict transportation plans to a given data manifold.
>
> **The practical side of BOT on manifolds.** Although much of the theory generalizes nicely to Riemannian manifolds, the generalization of the practical algorithms is highly non-trivial. Unlike in Euclidean space, realizing the optimal branching angles is a necessary but no longer sufficient condition for relatively optimal solutions. For instance, the meridians of three terminals located in the southern hemisphere at longitudes $0^\circ$, $120^\circ$, $240^\circ$, will intersect at both poles at angles of 120$^\circ$, which is the optimal angle for $\alpha=0$. Nonetheless, only the south pole is the optimal branching point. In essence, the geometry optimization aims to assign simultaneously to each branching point the coordinates of the weighted $L^1$ center of mass of its neighbors, a problem that is considered in [F, G]. The topology optimization step of our algorithm could be easily generalized if the geodesic distance can be computed. Due to these obstacles, previous works of the Steiner Tree problem ($\alpha=0$) have focused mostly on the sphere as important special case [H].
>
> **5 Structuring our theoretical and practical contributions (ooE5, CZsf)**
>
> Our paper comprises both theoretical and practical contributions which, in our mind, intertwine. For instance, the algorithm presented in Sect. 3, whose application is constrained to BOT in the two dimensional plane, establishes the theory for Sect. 4 (BOT properties) and 5 (BOT on manifolds). On the other hand, the studied theoretical BOT properties aid us in the construction of an efficient heuristic algorithm to approximate the optimal BOT solution (Sect. 6). We have considered splitting our paper (as suggested by reviewer ooE5) but we believe that the results presented in our work can be better understood together. Therefore, we have advocated for a self-contained manuscript which walks through the main results in the main paper, and provides the more technical details in the appendix. For the final version, we will try and make some of the technical results more accessible, and we will add more concrete references where needed.
>
> (Last not least we try our best to stem the bias towards smallest publishable units that can be observed occasionally.)
>
> _References_
> [E] Xin-yao, Jiang. The Steiner problem on a surface. (1987)
> [F] Afsari, Bijan. Riemannian $L^{p}$ center of mass: existence, uniqueness, and convexity. (2011)
> [G] Afsari, Bijan, Roberto Tron, and René Vidal. On the convergence of gradient descent for finding the Riemannian center of mass. (2013)
> [H] Dolan, John, Richard Weiss, and J. MacGregor Smith. Minimal length tree networks on the unit sphere. (1991)

---

> > ### Comment · Reviewer_CZsf · 2022-08-07
> > **Thank you for addressing my comments and questions**
> >
> > Thank you for addressing my comments and questions. I am satisfied with the answers and so I  would like to keep the score as it is.

---

> ### Author Response · Authors · 2022-08-02
> **Overarching Comments (2 of 3)**
>
> **2 Runtime of practical algorithms (mpG7, ooE5)**
>
> Though we provide information on the runtime of the numerical geometry optimization and greedy topology optimization (cf. Fig. 23 and Fig. 25 in the appendix), these were not discussed in the text. In the revised version we will amend this.
>
> As explained in Sect. 6.1, a single iteration of the geometry optimization has linear time complexity. However, the number of such iterations to optimize the geometry is not known from a theoretical point of view.
> Paralleling the setup of Fig. 23, we have now conducted an experiment where we report the number of iterations, for a batch of BOT problems. We performed the same experiment in Euclidean space of different dimensions ($d \in \\{3, 4, 5, 10,30 ,100\\} $) (see response to ooE5, paragraph 1 for a comment on the generalization to higher-dimensions BOT). Empirically, we found that independently of the convergence threshold $\\eta$ (see Alg. 1) and irrespective of the dimension of the BOT problem, the average number of iterations scales as $O(\\log(n))$. Given that a single iteration has complexity $O(nd)$, the full geometry optimization thus scales as $O(n \\log(nd))$, where $n$ is the number of terminals and $d$ the number of dimensions.
> Due to the theoretical focus of our work, we had not invested much time in optimizing our code. In particular, we used a standard solver to solve the system of equations in Eq. (7), which scaled worse than linearly (see appendix G.2). We have now developed an improved C++-implementation, which explicitly uses the linear elimination scheme, resulting in an order of magnitude speedup. We will make our improved implementation available together with the revised version of our manuscript.
>
>
> **3 Comparing our solver against existing approaches**
>
> - **Comparison against standard OT solvers (CZsf, mpG7)**: There exist efficient and also scalable approximate solvers (Sinkhorn-Knopp) [C, 5] for optimal transport (OT), i.e. BOT with $\\alpha=1$. On the contrary, BOT in general is NP-hard, and therefore our algorithm should not be seen as an alternative to OT solvers. Rather we can benefit from the existing fast solvers and build on them. For instance, for BOT problems with $\\alpha \\approx 1$, the OT solution provides a strong initial guess for the greedy topology optimization.
>
> - **Comparison against other BOT solvers (CZsf, ooE5)**: Unfortunately, the BOT solvers that have been proposed in the literature so far do not have code publicly available. We contacted the corresponding authors, but either did not get an answer or the code had not been maintained to be shared, with one notable exception: The author of [21] generously did share his code with us, but as mentioned in the beginning of Sect. 6, the algorithm requires some user supervision. Due to the large set of hand-crafted rules, this algorithm also requires a certain level of expertise. Still, we conducted experiments to the best of our ability, and png files with the results of the algorithm can be found in the supplemental material. Visually, these look worse than ours. Indeed, the algorithm generates unnecessary extra branching points with degree 2. Moreover, when $\\alpha=1$, i.e. the OT case, we observe that the algorithm still generates branches, while OT is characterized by the absence of branching. Finally, we stress the fact that for such a small example (6 terminals) the algorithm needs around 30 seconds, without counting the time invested in the supervision, while ours just takes a fraction of a second.
> The authors of [D] approach the BOT problem by phrasing it as a limit of functional minimization problems. Their algorithm discretizes the plane and the continuous cost function in order to approximate the optimal function solution. Thus, a fair comparison with our algorithm is not straightforward, since on the one hand our output is a graph while theirs is a discretized function. On the other hand, the cost they minimize is an approximation of the actual BOT cost. We have tried to reimplement their approach, but could not make the optimization converge.
> Consequently, comparison against other algorithms forces researchers to reimplement algorithms described in the literature, which may have partially hindered the more practical development of BOT. By making our code available we hope to aid the evolution of the field.
>
> _References_
> [C] Bonneel, N., Van De Panne, M., Paris, S., & Heidrich, W. Displacement interpolation using Lagrangian mass transport. (2011)
> [D] Oudet, Edouard and Santambrogio, Filippo. A Modica-Mortola approximation for branched transport and applications. (2011)

---

> > ### Comment · Reviewer_mpG7 · 2022-08-07
> > **Comparison against standard OT solvers.**
> >
> > Thank you for clarify the running time.
> > The authors mentioned that  "For instance, for BOT problems with $\alpha \approx 1$ , the OT solution provides a strong initial guess for the greedy topology optimization."
> > Would you please elaborate a bit on how does it work?  I.e. how does one use OT solution to help on election of  BP for BOT.

---

> > > ### Author Response · Authors · 2022-08-08
> > > **Optimal transport as input topology**
> > >
> > > Thank you for acknowledging our rebuttal.
> > > WLOG, the OT solution can be assumed to be acyclic [2]. As such, it provides a valid input topology for our greedy optimization (Alg. 3), even though it does not contain any branching points. Like in the case starting from a star graph input (see Fig. 24), the algorithm itself inserts (additional) branching points where beneficial/needed. Clearly, for $\alpha \approx 1$, the OT solution provides a reasonable starting point with relatively low cost (the objective function is almost the one of OT). From there, our algorithm may improve the solution only further by introducing branching points.

---

> > > > ### Comment · Reviewer_mpG7 · 2022-08-08
> > > > **Thank you for the quick reply**
> > > >
> > > > I see, thank you for the quick reply.

---

### Meta-Review · Area_Chair_kDRq · 2022-08-26

**Recommendation:** Accept
**Confidence:** Certain

**Metareview:**

The paper presents novel structural and algorithmic results for solving the branched optimal transport problem. In the problem, flow is to be routed from sources to sinks (terminals) in the plane with the possibility of adding non-terminal intermediate nodes. The flow cost on each edge is proportional to the distance between endpoints and subadditive in flow amount; this encourages solutions   with “branching”, where flow is routed along common paths. The problem is to select the topology of the graph, location of branching points, and flow amounts. The paper presents structural results about the optimal solution: it is always a tree, which also determines the optimal flow amounts. Results are also given about the branching factor and angles in an optimal solution, which are used in a heuristic algorithm for placing the branching points.

Reviewers unanimously found the results to be novel and interesting, and the paper of high quality. They appreciated the theoretical and algorithmic work. Reviewers questioned what applications this work might have (especially to ML) — no concrete applications were given in the paper, but the authors speculated on some in the rebuttal. Given this, two reviewers questioned whether the scope of the paper was a good match for NeurIPS. The meta-reviewer finds the match appropriate, given the interest in optimization and OT within the NeurIPS community, but these reviewer comments indicate that the audience is likely narrow, and the paper could be strengthened by connecting it to concrete applications.


**Award:**

No

---

### Decision · Program_Chairs · 2022-09-14

Accept